# JavisDiT: Joint Audio-Video Diffusion Transformer with Hierarchical Spatio-Temporal Prior Synchronization

**Kai Liu**[1,2]*, **Wei Li**[3]*, **Lai Chen**[1], **Shengqiong Wu**[2], **Yanhao Zheng**[1], **Jiayi Ji**[2],
**Fan Zhou**[1], **Jiebo Luo**[4], **Ziwei Liu**[5], **Hao Fei**[2]†, **Tat-Seng Chua**[2]

[1]Zhejiang University, [2]National University of Singapore,
[3]University of Science and Technology of China,
[4]University of Rochester, [5]Nanyang Technological University

## Abstract

This paper introduces `JavisDiT`, a novel Joint Audio-Video Diffusion Transformer designed for synchronized audio-video generation (JAVG). Based on the powerful Diffusion Transformer (DiT) architecture, JavisDiT simultaneously generates high-quality audio and video content from open-ended user prompts in a unified framework. To ensure audio-video synchronization, we introduce a fine-grained spatio-temporal alignment mechanism through a Hierarchical Spatial-Temporal Synchronized Prior (HiST-Sypo) Estimator. This module extracts both global and fine-grained spatio-temporal priors, guiding the synchronization between the visual and auditory components. Furthermore, we propose a new benchmark, `JavisBench`, which consists of 10,140 high-quality text-captioned sounding videos and focuses on synchronization evaluation in diverse and complex real-world scenarios. Further, we specifically devise a robust metric for measuring the synchrony between generated audio-video pairs in real-world content. Experimental results demonstrate that JavisDiT significantly outperforms existing methods by ensuring both high-quality generation and precise synchronization, setting a new standard for JAVG tasks. Our code, model, and data are available at https://javisverse.github.io/JavisDiT-page/.

## 1 Introduction

In the field of AI-generated content (AIGC), multimodal generation that covers images, videos, and audio has gained increasing attention (Lyu et al., 2024; Liu et al., 2024b; Yang et al., 2024b; Lyu et al., 2025), where diffusion-based models (Ho et al., 2020; Liu et al., 2023b; Peebles & Xie, 2023) have demonstrated remarkable performance. While early studies mainly focused on single-modality generation, recent work has shifted toward the simultaneous generation of multiple modalities (Gadre et al., 2024; Wu et al.). Notably, synchronized audio and video generation (Ruan et al., 2023; Xing et al., 2024; Sun et al., 2024) has emerged as a crucial area of study. Audio and video are inherently interconnected in most real-world scenarios, making their joint generation highly valuable for applications, such as movie production and short video creation. Current approaches to synchronized audio-video generation can be broadly categorized into two types. The first involves asynchronous pipelines, where audio is generated first and then used to synthesize video (Yariv et al., 2024; Jeong et al., 2023; Zhang et al., 2025), or vice versa (Comunità et al., 2024; Zhang et al., 2024; Xie et al., 2024). The second type involves end-to-end Joint Audio-Video Generation (namely JAVG) (Ruan et al., 2023; Xing et al., 2024; Sun et al., 2024), which, by avoiding noise accumulation during cascading processes of the first type, has thus attracted more research attention. Overall, it is strongly and widely believed that promoting JAVG requires two equally critical criteria: (1) *ensuring high-quality generation of audio and video*, and (2) *pursuing precise synchronization between two modalities*.

---

*Equal contribution. Work done during Kai Liu's visiting period at NUS. Email: kail@zju.edu.cn.
†Corresponding author. Email: haofei7419@gmail.com

On one hand, for high-quality audio-video generation, the backbone modules of both branches need to be as strong as possible. Recent JAVG research (Ruan et al., 2023; Xing et al., 2024; Sun et al., 2024) attempts to take Diffusion Transformer (DiT) (Peebles & Xie, 2023) as backbone architectures, due to the remarkably enhanced performance for vision and even audio tasks. In particular, AV-DiT (Wang et al., 2024b) and MM-LDM (Sun et al., 2024) inherit an image DiT (Peebles & Xie, 2023; Esser et al., 2024) to generate video and audio, where the fine-grained spatio-temporal modeling capability can be however limited. Concurrent works like Uniform (Zhao et al., 2025) and SyncFlow (Liu et al., 2024a) take the enhanced STDiT3 (Zheng et al., 2024) blocks, but they either simply concatenate video/audio tokens or utilize a mono-directional video-to-audio adapter, lacking sufficient mutual information exchange between the two channels.

On the other hand, in terms of synchronization between the audio and video channels in JAVG, extensive investigations have been made to explore more effective alignment strategies. However, most of these works still fall prey to insufficient modeling of synchronization. For example, some emphasize coarse-grained temporal alignment by implementing straightforward parameter sharing (Ruan et al., 2023; Wang et al., 2024b; Zhao et al., 2025) or temporal modulating (Wang et al., 2024b; Liu et al., 2024a) between the audio and video channels. Others focus on coarse-grained semantic alignment, such as Seeing-Hearing (Xing et al., 2024), which carries out simple audio-video embedding alignment, and MM-LDM (Sun et al., 2024) performing straightforward representation align-

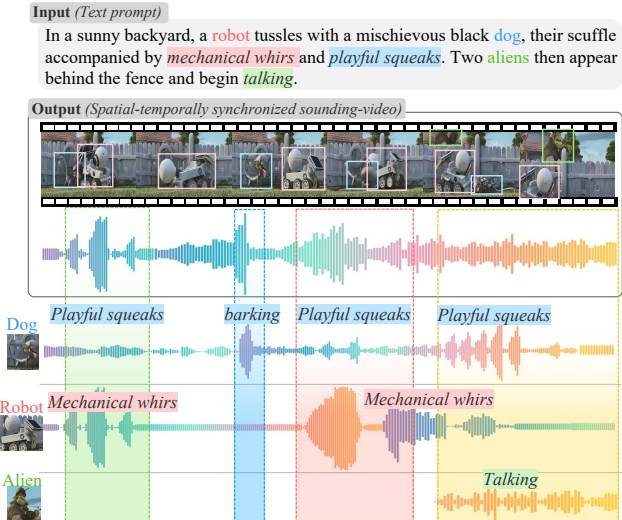

Figure 1: Given the text prompt, a JAVG system generates a spatial-temporally synchronized sounding video. The sounds align precisely with the temporal progression of the actions.

ment in the latent space of the diffusion model. These approaches fail to consider a *fine-grained spatio-temporal alignment*, which is essential for realistic synchronized audio-video generation. Specifically, in a realistic-sounding video, all visual and auditory contents should be dictated by spatio-temporal characteristics of various objects, *i.e.*, **a)** spatially distinct visual content (*e.g.*, where events occur and how objects move), and **b)** temporally corresponding auditory attributes, such as source category and synchronized duration. We illustrate this in Fig. 1: A dog and a robot are playing on the ground, while an alien appears and starts to talk. In this scene, the robot's mechanical whirs and the dog's squeaks and barks keep consistent, and the alien's talking emerges later and continues to the end. Overall, the visual content and audio stay synchronized both temporally and spatially.

To address the above dual challenges, this paper proposes a novel niche-targeting JAVG system, called **JavisDiT** (*cf.* Fig. 2). First, JavisDiT adopts the DiT architecture as the backbone, where the audio and video channels share AV-DiT blocks to enable high-quality audio-video generation. Within JavisDiT, we design three infrastructural blocks: Spatio-Temporal Self-Attention, Coarse-Grained Cross-Attention, and Fine-Grained Spatio-Temporal Cross-Attention (ST-CorssAttn). To implement the fine-grained spatio-temporal alignment idea, we design a Hierarchical Spatial-Temporal Synchronized Prior (*HiST-Sypo*) estimation module. The HiST-Sypo Estimator hierarchically extracts the following from the input conditional prompt: ● *global coarse-grained spatio-temporal priors*, such as the semantic framework of the overall sounding video; ● *fine-grained spatio-temporal priors*, such as the distinct visual content triggered by various sounds and their corresponding temporal priors.

These global and fine-grained priors serve as spatio-temporal features, injected into different AV-DiT blocks to guide the spatial semantic and temporal synchronization of both audio and video. Built on JavisDiT, we further design a contrastive learning-based (Chen et al., 2020b) HiST-Sypo estimation strategy to learn robust spatial-temporal synchronized prior knowledge from large-scale data of sounding videos.

Further, we observe that the existing JAVG evaluation benchmarks, *e.g.*, Landscape (Lee et al., 2022) and AIST++ (Li et al., 2021) suffer from certain limitations, including overly simplistic audio-video content and limited scene diversity (Mao et al., 2024). This creates an evident gap compared to the complex audio-video content encountered in real-world applications, *i.e.*, models (Ruan et al., 2023; Sun et al., 2024) trained there largely suffer from out-of-domain issue, thereby partially hindering the development and practical applicability on open environment. To bridge these gaps, we introduce a new JAVG benchmark, called `JavisBench`. After rigorous manual inspection, we collect a total of 10,140 high-quality text-captioned sounding videos, which span over 5 dimensions with 19 scene categories. Notably, more than 50% of these videos feature highly complex and challenging scenarios, making the dataset more representative of diverse real-world applications. Meanwhile, we find that existing JAVG evaluation method can also be limited by failing to adequately assess the JAVG systems on complex sounding videos. Thus, we devise a novel metric (JavisScore) based on a temporal-aware semantic alignment mechanism.

Extensive experiments on both existing JAVG benchmarks and our JavisBench dataset demonstrate that JavisDiT significantly outperforms state-of-the-art methods in generating high-quality sounding videos. Our system is especially effective in handling complex-scene videos, thanks to the HiST-Sypo estimation mechanism. In summary, our main contributions are as follows:

- We introduce `JavisDiT`, a novel JAVG model with a hierarchical spatio-temporal prior estimation mechanism to achieve audio-video synchronization in both spatial and temporal dimensions.
- We contribute `JavisBench`, a new large-scale JAVG benchmark dataset with challenging scenarios, along with robust metrics to evaluate audio-video synchronization, offering a comprehensive baseline for future research.
- Empirically, our system achieves state-of-the-art performance across both existing closed-set and our open-world benchmarks, setting a new standard for JAVG.

## 2 PRELIMINARIES

**Task Definition and Formulation.** Joint Audio-Video Generation (JAVG) requires diffusion models $\mathcal{G}_\theta$ to simultaneously generate videos $\mathbf{v}$ and corresponding audios $\mathbf{a}$ given a text input $s$. The forward process at timestamp $t$, either with DDPM (Ho et al., 2020) or FlowMatching (Lipman et al., 2023), is formulated as:

$$(\mathbf{v}_{t-1}, \mathbf{a}_{t-1}) = \mathcal{G}_\theta(\mathbf{v}_t, \mathbf{a}_t, s, t). \tag{1}$$

$\mathbf{v} \in \mathbb{R}^{T_v \times (H \times W) \times 3}$ denotes a video with $T_v$ frames and 3 RGB channels, where $H, W$ are the frame height and width. On the other hand, $\mathbf{a} \in \mathbb{R}^{T_a \times M \times 1}$ encodes audio in the image-like mel-spectrogram with $T_a$ temporal frames and 1 channel, where $M$ is the frequency bins.

**Spatio-Temporal Alignment.** For an ideal model $\mathcal{G}$, the generated video and audio should maintain coarse-grained semantic alignment with the input text $s$, while further achieving fine-grained spatio-temporal alignment between video $\mathbf{v}$ and audio $\mathbf{a}$, which are formally defined as:

- *Spatial Alignment* means the occurrence of a specific event in a region of the video frame (at the $H \times W$ dimension in $\mathbf{v}$), which corresponds to the appearance of matching frequency components in the audio (at the $M$ dimension in $\mathbf{a}$).
- *Temporal Alignment* means the onset or termination of an event at a specific frame or timestamp in the video (at the $T_v$ dimension in $\mathbf{v}$), which must coincide with the corresponding start or stop of the response in the audio (at the $T_a$ dimension in $\mathbf{a}$).

We focus on the joint generation of high-quality and spatio-temporally aligned video-audio pairs.

## 3 THE PROPOSED JAVISDIT SYSTEM

### 3.1 JAVISDIT MODEL ARCHITECTURE

Fig. 2 illustrates the overall architecture. Spatio-temporal attention is employed for effective cross-modal alignment while ensuring high-quality generation for videos and audios. Each branch uses *ST-SelfAttn* for intra-modal aggregation, incorporates coarse text semantics via *CrossAttn*, integrates fine-grained spatio-temporal priors through *ST-CrossAttn*, and enhances video-audio fusion with *Bi-CrossAttn*. Here we introduce the core components displayed in Fig. 2 (b):

**Spatio-Temporal Self-Attention.** Given that both videos and audios possess spatial and temporal attributes (Ruan et al., 2023), we employ a cascaded spatio-temporal self-attention mechanism for

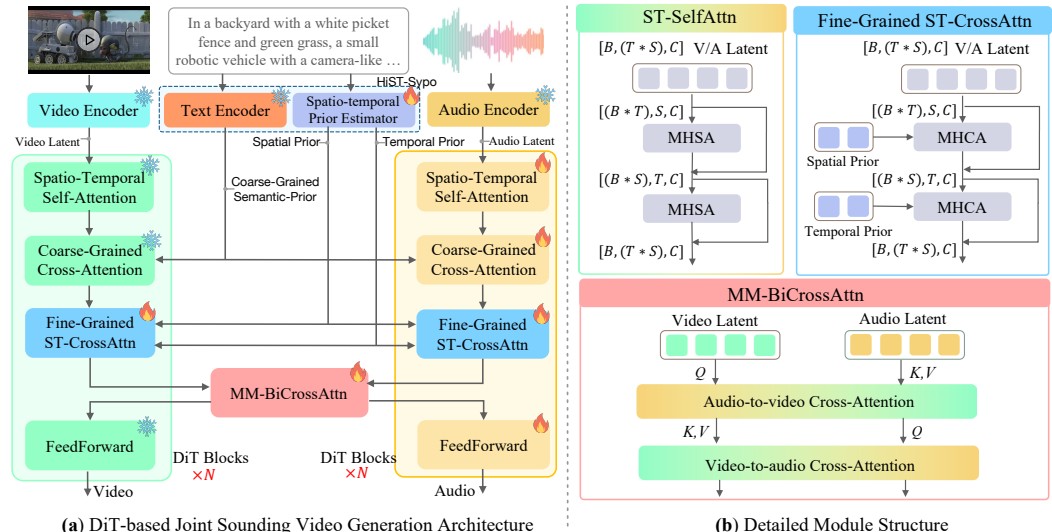

Figure 2: On the left, we present the overall DiT-based sounding video generation architecture of **JavisDiT**, consisting of a video generation branch, audio generation branch, **HiST-Sypo Estimator** module, and the **MM-BiCrossAttn** module. On the right, we illustrate the detailed structural design of Spatio-temporal Self-attention (**ST-SelfAttn**), Fine-grained Spatio-temporal Cross-attention (**Fine-Grained ST-CrossAttn**), and Multi-Modality Bidirectional Cross-attention (**MM-BiCrossAttn**).

intra-modal information aggregation. As illustrated Fig. 2(b), MHSA is applied sequentially along the spatial $((H \times W)$ for $\mathbf{v}$, $M$ for $\mathbf{a})$ and temporal ($T_v$ for $\mathbf{v}$, $T_a$ for $\mathbf{a}$) dimensions. This efficiently achieves fine-grained spatio-temporal modeling with reduced computational cost.

**Spatio-Temporal Cross-Attention.** For a text prompt $s$, after injecting coarse semantics from the T5 encoder (Raffel et al., 2020) via vanilla cross attention, we estimate spatial and temporal priors with $N_s$ and $N_t$ learnable $c$-dimensional tokens by our ST-Prior Estimator (see details in Sec. 3.2). As Fig. 2(b) shows, spatial and temporal priors guide cross-attention in both branches along the spatial and temporal dimensions, enabling unified, fine-grained conditioning for video-audio synchronization.

**Cross-Modality Bidirectional-Attention.** After aligning video and audio with ST-Prior, we incorporate a bidirectional attention module (Liu et al., 2025) to enable direct cross-modal interactions. As Fig. 2(b) illustrates, after computing the attention matrix $A$ between $q_v$ and $k_a$, we first multiply $A$ with $v_a$ to obtain the audio-to-video cross-attention. Similarly, multiplying $A^T$ (the transpose of $A$) with $v_v$ yields the video-to-audio cross-attention. This mechanism enhances cross-modal information aggregation to facilitate high-quality joint audio-video generation.

## 3.2 HIERARCHICAL SPATIAL-TEMPORAL SYNCHRONIZED PRIOR ESTIMATOR

Unlike previous works, our JavisDiT derives two hierarchical conditions (priors) from texts: a global semantic prior for coarse event suggestion (*what*) and a fine-grained spatio-temporal prior (ST-Prior) to specify event timing and location (*when* and *where*). These two complementary priors are unified into a HiST-Sypo Estimator, enabling precise synchronization between generated video and audio.

**Coarse-Grained Spatio-Temporal Prior.** Since the default semantic embeddings from T5 encoder (Raffel et al., 2020) are strong enough to coarsely describe the overall sounding event, we simply reuse T5 embeddings as our coarse-grained spatio-temporal prior (or, semantic prior).

**Fine-Grained Spatio-Temporal Prior.** Text inputs typically provide coarse-grained descriptions of events, *e.g.*, "a car starts its engine and leaves the screen". To enable fine-grained conditioning, we estimate spatio-temporal priors: *spatial prior* specifies where events occur (*e.g.*, "the car is in the top-left corner of screen"), and *temporal prior* defines when they start and end (*e.g.*, "sound starts at 2s, exits at 7s, fades at 9s"). Instead of generating explicit prompts from LLMs (Liu et al., 2024b), we efficiently estimate spatio-temporal priors as latent token conditions to guide the diffusion process.

**Fine-Grained ST-Prior Estimation.** As shown in Fig. 3, for a given text input $s$, we utilize the 77 hidden states from ImageBind's (Girdhar et al., 2023) text encoder, and use $N_s = 32$ spatial tokens $\mathbf{p}_s$ and $N_t = 32$ temporal tokens $\mathbf{p}_t$ to query a 4-layer transformer $\mathcal{P}$ to extract spatio-temporal information. Since input text prompts usually may not specify an event's happening time and location, it may almost occur and cease in arbitrary location and timing, and the same text $s$ should correspondingly yield different ST-Priors $(\mathbf{p}_s, \mathbf{p}_t)$. To capture this variability, our ST-Prior Estimator $\mathcal{P}$ outputs the mean and variance of a Gaussian distribution, from which a plausible $(\mathbf{p}_s, \mathbf{p}_t)$ is sampled: $(\mathbf{p}_s, \mathbf{p}_t) \leftarrow \mathcal{P}_\phi(s; \epsilon)$, where $\epsilon$ is a Gaussian noise. Furthermore, we carefully devised a contrastive learning approach to learn a robust ST-Prior Estimator, which involves a series of

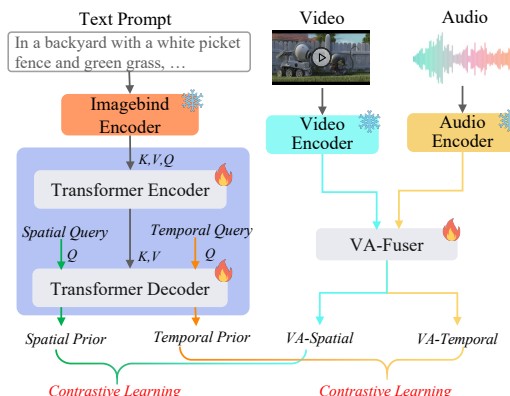

Figure 3: The framework of **Spatio-temporal Prior Estimator** with a 4-layer transformer encoder-decoder (referring to the purple region). Contrastive learning is utilized for optimization.

negative sample (asynchronous video-audio pairs) construction strategies and specifically-designed loss functions. Implementation details are provided in the Sec. C.2.

### 3.3 MULTI-STAGE TRAINING STRATEGY

Our JavisDiT focuses on two core objectives: achieving high-quality single-modal generation and ensuring fine-grained spatio-temporal alignment between generated videos and audios. To do this, we leverage the pretrained weights of OpenSora (Zheng et al., 2024) for the video branch, and adopt a three-stage training strategy for robust joint video-audio generation:

1) **Audio Pretraining**. We initialize the audio branch with OpenSora's video branch weights and train it on 0.8M audio-text samples to ensure superior single-modal generation quality.
2) **ST-Prior Training**. We train the ST-Prior Estimator $\mathcal{P}_\phi$ using 0.6M synchronous text-video-audio triplets and synthesized asynchronous negative samples to strengthen representation.
3) **JAVG Training**. We freeze the video and audio branches' self-attention blocks and ST-Prior Estimator, training only the ST-CrossAttn and Bi-CrossAttn modules with 0.6M samples to enable synchronized video-audio generation.

In addition, dynamic temporal masking (Zheng et al., 2024) enables flexible adaptation of JavisDiT to x-conditional tasks (v2a/a2v generation, image animation, video-audio extension, *etc.*). Details refer to Sec. E.7.

## 4 A CHALLENGING JAVISBENCH BENCHMARK

A strong generative model must ensure diverse video content, audio types, and fine-grained spatio-temporal synchrony. However, current JAVG evaluation benchmarks fall short in encompassing the diversity of scenes, the complexity of real-world environments, and the presence of multiple sounding events within a single audio-video instance, thereby constraining comprehensive and realistic evaluation (detailed in Sec. D.1). To combat this, we propose a more challenging benchmark featuring complex multi-event video-audio pairs (Sec. 4.1), along with a robust synchronization-oriented metric tailored for evaluating fine-grained spatio-temporal alignment (Sec. 4.2).

### 4.1 DATA CONSTRUCTION

**Taxonomy.** We design five evaluation dimensions from coarse to fine:

- *Event Scenario* depicts scenarios where audio-visual events happen, such as nature or industry.
- *Video Style* describes the visual style of given videos, such as camera shooting or 2D animations.
- *Sound Type*: the sounding type of given audios, such as sound effects or music.
- *Spatial Composition* defines the sounding subjects appearing in videos and audios, divided by single or multiple sounding subjects that exist.
- *Temporal Composition* describes events' onsets and terminations in v-a pairs, distinguished by single or multiple sounding sources that sequentially or concurrently occur.

| Benchmark | #Sample | Scenarios | Categories |
|---|---|---|---|
| AIST++ | 20 | 1 (dancing) | 1 (music) |
| Landscape | 100 | 1 (natural) | 1 (ambient) |
| TAVGBench | 3,000 | Unknown | Indiscriminate |
| **Ours** | 10,140 | Diversified | Hierarchical |

Figure 4: **Left**: Comparison between JavisBench and other benchmarks. TAVGBench (Mao et al., 2024)'s test set is not yet released (Unknown). **Right**: Detailed category distribution of JavisBench.

Using GPT-4 (Achiam et al., 2023), we develop a hierarchical categorization system with 5 dimensions and 19 categories (Fig. 4), and the detailed definitions are presented in Sec. D.2.

**Data Curation.** There are two data sources: (1) test sets of existing datasets (*e.g.*, Landscape/AIST++ and FAVDBench (Shen et al., 2023)), and (2) YouTube videos uploaded between June and December 2024 to prevent data leakage (Mao et al., 2024). To collect YouTube videos, we prompt GPT-4 to generate category-specific keywords using our defined taxonomy, enabling efficient and targeted video collection while avoiding noisy data. After strict manual legal and ethical verification (see Sec. A.1), the above process yielded around 30K sounding video candidates. With several filtering tools to ensure quality and diversity, we use the advanced Qwen-family models (Yang et al., 2024a; Wang et al., 2024c; Chu et al., 2024) to generate captions and categorize video-audio pairs into desired taxonomy, and finally curate 10,140 diverse and high-quality video-audio pairs for JavisBench-10K. More details about data construction are presented in Sec. D.3. We also randomly select 1,000 samples to form the version of JavisBench-mini for efficient evaluation.

**Benchmark Statistics.** Fig. 4 highlights JavisBench's contributions: (1) offering more diverse data compared to AIST++ (Li et al., 2021) and Landscape (Lee et al., 2022), (2) providing a detailed taxonomy for comprehensive evaluations, surpassing TAVGBench (Mao et al., 2024), and (3) featuring the first evaluation benchmark focused on multi-event synchronization. As Fig. 4 shows, JavisBench covers diverse *event scenarios*, *visual styles*, and *audio types*, ensuring a balanced distribution for greater **diversity**. Most realistic but under-represented scenarios like 2D/3D animations (25%) and industrial events (13%) are included. Furthermore, JavisBench emphasizes spatial and temporal complexity. 75% of samples feature *multiple sounding events*, 28% involve *sequential events*, and 57% include *simultaneous events*. These diverse multi-event scenarios pose significant **challenges** for JAVG models, particularly in achieving accurate spatial and temporal alignment.

## 4.2 JAVISSCORE: A MORE ROBUST JAVG METRIC

**Motivation.** AV-Align (Yariv et al., 2024) is a widely-adopted JAVG metric, which uses video optical-flow estimation to match the onset detected in audios to measure temporal synchronization. However, AV-Align may struggle with complex scenarios (*i.e.*, with multiple sounding events or subtle visual movements) and produce misleading results (see Sec. D.4). Therefore, we propose a more robust evaluation metric, namely JavisScore, to measure spatiotemporal synchronization in diverse, real-world contexts.

**Implementation and Verification.** Technically, we chunk each video-audio pair into several clips with 2-second window size and 1.5-second overlap, use ImageBind (Girdhar et al., 2023) to compute the audio-visual synchronization for each segment, and average the scores as the final metric:

$$S_{Javis} = \frac{1}{W} \sum_{i=1}^{W} \sigma\left(E_v(V_i), E_a(A_i)\right), \tag{2}$$

where $E_v$ and $E_a$ are vision and audio encoders, $A_i$ and $V_i$ are audio and video segments in $i$-th window, $W$ is the number of windows, and $\sigma$ is a specifically designed synchronization measure for video-audio segments (see Sec. D.4). Motivated by Mao et al. (2024), we calculate similarities between all frames and the audio within each segment, and select the 40% least synchronized frames to obtain the score of the current window. Sec. D.4 also presents a human-annotated evaluation dataset with 3,000 samples to verify JavisScore's efficacy against previous metrics.

Table 1: **Main results on proposed JavisBench.** Models are evaluated with 4-second videos at 240P/24fps and audio at 16kHz. JavisDiT comprehensively outperforms or gets on par with currently available SoTA approaches. **Best** and secondary results are marked **bold** and underline respectively.

| Method | AV-Quality | | | Text-Consistency | | | | AV-Consistency | | | AV-Synchrony |
|---|---|---|---|---|---|---|---|---|---|---|---|
| | FVD↓ | KVD↓ | FAD↓ | TV-IB↑ | TA-IB↑ | CLIP↑ | CLAP↑ | AV-IB↑ | CAVP↑ | AVHScore↑ | JavisScore ↑ |
| *- T2A+A2V* | | | | | | | | | | | |
| TempoTkn (Yariv et al., 2024) | 539.8 | 7.2 | - | 0.084 | - | 0.205 | - | 0.137 | 0.787 | 0.122 | 0.103 |
| TPoS (Jeong et al., 2023) | 839.7 | 4.7 | - | 0.201 | - | 0.229 | - | 0.142 | 0.778 | 0.129 | 0.095 |
| *- T2V+V2A* | | | | | | | | | | | |
| ReWaS (Jeong et al., 2024) | - | - | 9.4 | - | 0.123 | - | 0.280 | 0.110 | 0.794 | 0.104 | 0.079 |
| See&Hear (Xing et al., 2024) | - | - | 7.6 | - | 0.129 | - | 0.263 | 0.160 | 0.798 | 0.143 | 0.112 |
| FoleyCftr (Zhang et al., 2024) | - | - | 9.1 | - | **0.149** | - | 0.383 | 0.193 | 0.800 | **0.186** | 0.151 |
| *- T2AV* | | | | | | | | | | | |
| MM-Diff (Ruan et al., 2023) | 2311.9 | 12.2 | 27.5 | 0.080 | 0.014 | 0.181 | 0.079 | 0.119 | 0.783 | 0.109 | 0.070 |
| UniVerse-1 (Wang et al., 2025) | **194.2** | **0.5** | 8.7 | **0.272** | 0.111 | **0.309** | 0.245 | 0.104 | 0.793 | 0.098 | 0.077 |
| **JavisDiT**(Ours) | 204.1 | 1.4 | **7.2** | 0.263 | 0.143 | 0.302 | **0.391** | **0.197** | **0.801** | 0.179 | **0.154** |

Table 2: **Experimental results on previous datasets.** Numbers are borrowed from their released papers, and our model still performs best.

| Method | Landscape | | | AIST++ | | |
|---|---|---|---|---|---|---|
| | FVD↓ | KVD↓ | FAD↓ | FVD↓ | KVD↓ | FAD↓ |
| MM-Diff | 332.1 | 26.6 | 9.9 | 219.6 | 49.1 | 12.3 |
| See&Hear | 326.2 | 9.2 | 12.7 | - | - | - |
| AV-DiT | 172.7 | 15.4 | 11.2 | **68.8** | 21.0 | 10.2 |
| MM-LDM | 105.0 | 8.3 | 9.1 | 105.0 | 27.9 | 10.2 |
| **JavisDiT**(Ours) | **94.2** | **7.8** | **8.5** | 86.7 | **19.8** | **9.6** |

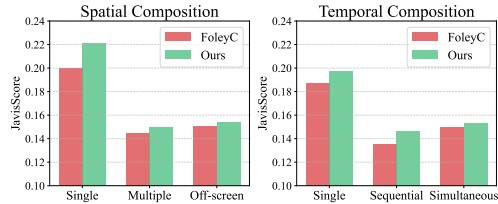

Figure 5: **Generative av-synchrony with Javis-Bench's taxonomy.** Current SOTA models still suffer from challenging scenarios.

# 5 EXPERIMENTS

## 5.1 SETUP

**Evaluation Datasets and Metrics.** The proposed JavisBench is used as the primary benchmark for comprehensive evaluation. Multiple metrics are reported: (1) audio/video quality, (2) semantic consistency against conditional texts, (3) semantic consistency and (4) spatio-temporal synchrony between videos and audios. We also evaluate on AIST++ (Li et al., 2021) and landscape (Lee et al., 2022) for consistency, and follow prior works (Ruan et al., 2023; Sun et al., 2024) to report FVD, KVD, and FAD.

**Compared Methods.** For JavisBench, we reproduce and compare a series of baselines: (1) for cascaded T2A+A2V methods (Yariv et al., 2024; Jeong et al., 2023), we use AudioLDM2 (Liu et al., 2024b) for the prepositive T2A task; (2) for cascaded T2V+V2A methods (Jeong et al., 2024; Xing et al., 2024; Zhang et al., 2024), OpenSora (Zheng et al., 2024) is adopted to generate videos at first; and (3) for JAVG models, we currently compare with MM-Diffusion (Ruan et al., 2023) and UniVerse-1 Wang et al. (2025) since the others (Liu et al., 2024a; Zhao et al., 2025) are not currently open-sourced. For AIST++ and Landscape, we directly adopt the reported results.

**Implementation Details.** We collect 780K audio-text pairs for audio pertaining by 55 epochs, with 610K video-audio-text triplets to train ST-Prior Estimator for 1 epoch and JavisDiT for 2 epochs. The learning rate is 1e-5 for ST-Prior Estimator and 1e-4 for DiT. The video encoder-decoder and audio encoder-decoder are taken from OpenSora (Zheng et al., 2024) and AudioLDM2 (Liu et al., 2024b) and kept frozen. Detailed configurations refer to Sec. C.1.

## 5.2 MAIN RESULTS AND OBSERVATIONS

**Our JavisDiT achieves superior single-Modal quality and video-audio synchrony.** As shown in Tab. 1, our carefully designed spatiotemporal DiT architecture demonstrates exceptional unimodal generation quality, achieving significantly superior results compared to UNet-based architectures (*e.g.*, TempoToken (Yariv et al., 2024)) and naive DiT architectures (*e.g.*, MM-Diffusion (Ruan et al.,

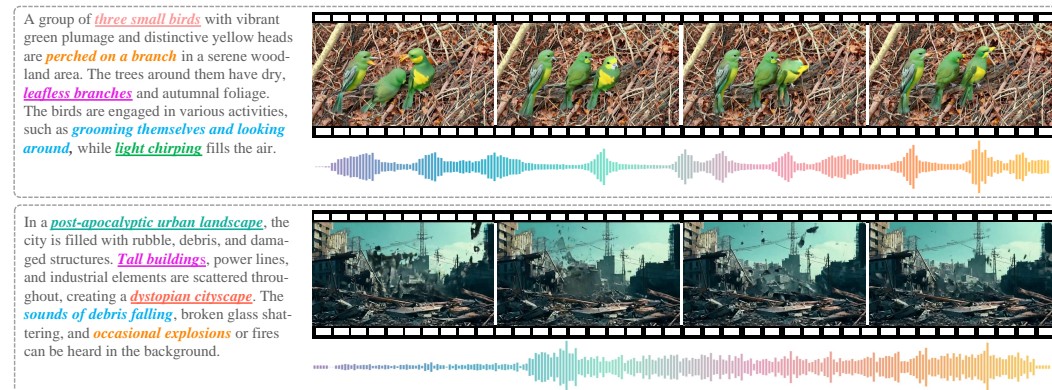

Figure 6: JavisDiT precisely captures the visual and auditory clues from text inputs to generate faithful sounding-videos with high-quality spatio-temporal alignments. Colored texts are spatio-temporal objects (underlined) and actions. More cases are shown in Sec. E.6.

2023)), with remarkable FVD (204.1) and FAD (7.2) scores. Meanwhile, from the perspective of global semantic alignment, including text-consistency and video-audio consistency, our model also achieves state-of-the-art performance, as evidenced by its TA-IB score of 0.263 and CLIP similarity score of 0.302. We attribute the gap between our JavisDiT and UniVerse-1 (Wang et al., 2025) in video-text alignment to the gap of pretrained models (we use OpenSora while UniVerse-1 takes more powerful Wan2.1 (Wan et al., 2025)). Fig. 6 showcases some representative JAVG examples. Notably, in terms of audio-video synchrony, our end-to-end model outperforms various cascaded and joint audio-video generation approaches, achieving a JavisScore of 0.154, surpassing the state-of-the-art cascaded method FoleyCrafter (Zhang et al., 2024).

To ensure a rigorous comparison, we follow the standard settings from prior works (Ruan et al., 2023; Xing et al., 2024; Sun et al., 2024) and train our model for 300 epochs on two closed-set datasets, including Landscape (Lee et al., 2022) and AIST++ (Li et al., 2021). As demonstrated in Tab. 2, our method consistently achieves state-of-the-art performance, with FVD of 94.2 on Landscape and FAD of 9.6 on AIST++. These results further highlight the superiority of our meticulously designed DiT architecture and hierarchical spatial-temporal prior estimator.

**Current models fail to simulate complex scenarios.** Fig. 5 presents FoleyCrafter(Zhang et al., 2024) and our JavisDiT across JavisBench categories, showing that existing models — including ours — struggle with AV synchrony in complex scenarios. When the sounding video contains only a single sounding object (*e.g.*, a person playing the violin alone), JavisScore is generally higher than in multi-object cases (*e.g.*, a street performance with multiple instruments), as the latter requires identifying the correct visual-audio correspondence. Similarly, videos with multiple simultaneous events (*e.g.*, a dog barking while a car horn sounds) yield lower JavisScore than single-event cases (*e.g.*, a person clapping), due to the increased challenge of modeling event timing and interactions. Sec. D.5 and Fig. A8 thoroughly highlight the limitations in handling real-world complexity.

## 5.3 IN-DEPTH ANALYSES AND DISCUSSIONS

To efficiently evaluate our proposed method, we perform the 3rd stage (JAVG) training using a subset of 60K entries from our entire training data, and test the models on JavisBench-mini (with 1,000 randomly selected samples from JavisBench). For simplicity, we report three normalized scores for a clear comparison from three dimensions: (1) **Quality**: $S_{\text{AVQ}} = 0.01 \times S_{\text{FVD}} + S_{\text{KFD}} + 0.1 \times S_{\text{FAD}}$; (2) **Consistency**: $S_{\text{AVC}} = S_{\text{AV-IB}} + S_{\text{CAVP}} + S_{\text{AVHScore}}$; (3) **Synchrony**: $S_{\text{AVS}} = S_{\text{JavisScore}}$.

**Well-designed DiT backbone is superior.** In Tab. 3, we first build a vanilla baseline to replace the spatio-temporal transformer backbone by UNet (Liu et al., 2024b) for the audio branch, and the generation receives a pool AV-Quality at 9.371. After replacing the UNet backbone with STDiT modules (Zheng et al., 2024), $S_{\text{AVQ}}$ immediately improves to 7.293, demonstrating the efficacy of the STDiT architecture. Then, we respectively incorporate the HiST-Sypo estimator and the bidirectional cross-attention (BiCA) modules for video-audio information sharing. Accordingly, HiST-Sypo brings significant enhancement on AV-Consistency (1.191 *vs.* 1.155) and AV-Synchrony

Table 3: **Ablation on the model design.** The specifically developed DiT architecture and the HiST-Sypo estimator jointly contribute to single-modal quality and va-synchrony. Full evaluation results are presented in Tab. A6.

| STDiT | HiST-Sypo | BiCA | Quality↓ | Consist↑ | Sync↑ |
|---|---|---|---|---|---|
| × | × | × | 9.371 | 1.140 | 0.118 |
| √ | × | × | 7.293 | 1.155 | 0.130 |
| √ | √ | × | 6.127 | 1.191 | 0.150 |
| √ | × | √ | 6.581 | 1.157 | 0.133 |
| √ | √ | √ | **6.012** | **1.201** | **0.153** |

Table 4: **Ablation on token number and injection strategies of ST-Priors.** We provide more experimental results in Fig. A9 and Tab. A9. Full evaluation results are presented in Tab. A7.

| $N_s$ | $N_t$ | Injection | Quality↓ | Consist↑ | Sync↑ |
|---|---|---|---|---|---|
| 0 | 0 | - | 6.581 | 1.157 | 0.133 |
| 1 | 1 | CrossAttn | 6.909 | 1.188 | 0.137 |
| 16 | 16 | CrossAttn | 6.322 | 1.200 | 0.151 |
| 32 | 32 | CrossAttn | **6.012** | **1.201** | **0.153** |
| 32 | 32 | Addition | 6.267 | 1.183 | 0.144 |
| 32 | 32 | Modulate | 6.190 | 1.191 | 0.145 |

(0.150 *vs.* 0.130), more than the simple BiCA module (1.157 *vs.* 1.155 for $S_{AVC}$ and 0.133 *vs.* 0.130 for $S_{AVS}$). This verifies our motivation that a simple channel-sharing mechanism in AV-DiT (Wang et al., 2024b) cannot effectively build audio-video synchronization, which can be achieved by our proposed fine-grained ST-Prior guidance instead. After bridging STDiT, HiST-Sypo, and BiCA modules, our JavisDiT reaches the best performance for both single-modal quality ($S_{AVQ} = 6.012$) and video-audio synchronization ($S_{AVC} = 1.201$ and $S_{AVS} = 0.153$), demonstrating the superiority of the well-designed DiT backbone.

**Spatio-temporal prior is effective and generic.** In Tab. 4, we take a preliminary step to investigate the number of spatio-temporal priors and the way to inject ST-Priors for better video-audio synchronization. We first take the model without ST-CA modules (the 4th row in Tab. 3) as our baseline (the 1st row in Tab. 4), and add ST-CA modules by gradually increasing the prior number from 1 to 32. According to the results, both single-modal quality ($S_{AVQ}$) and video-audio synchrony ($S_{AVC}$ and $S_{AVS}$) consistently increase as the prior number gains. Then, we try to utilize the 32 spatial-temporal priors for addition (adding to video/audio latent representations like a conditional embedding) and modulation (mapping priors to scales and biases to modulate video/audio representations). Although the final performance is inferior to the utilization of cross-attention, ST-priors still considerably enhance all metrics (*e.g.*, 0.144/0.145 *vs.* 0.133 for $S_{AVS}$). Besides, we further experiment and evaluate the influence of prior numbers and dimensions in Fig. A9, and verify the optimization objectives during ST-priors' estimation in Tab. A9. All the empirical results verify the effectiveness and versatility of our estimated spatio-temporal priors.

**How does ST-Prior ensure synchronized video&audio?** Fig. 7 illustrates the synchronization mechanism for our ST-priors to guide the video-audio generation process. In particular, we visualize the spatial-temporal cross-attention map from the last block in JavisDiT at the last sampling step on both video and audio branches. The qualitative results in Fig. 7 show the spatial priors successfully help JavisDiT focus on the subject that would produce the sound (in this case, it is the bubbles rather than the diver that can make the sound), and the temporal priors bring nearly uniform attention scores along the timeline (as the

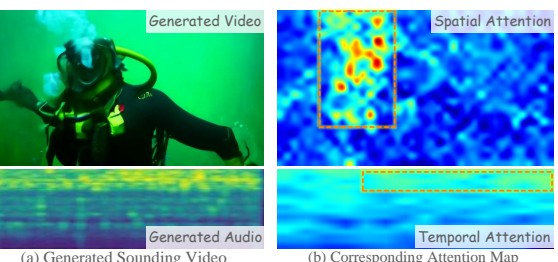

(a) Generated Sounding Video    (b) Corresponding Attention Map

Figure 7: **Visualization of cross-attention maps by spatio-temporal priors.** Spatial priors successfully capture the sounding subjects (bubbles in this case), and temporal priors accurately cover the whole timeline for the continuous sounding event.

bubbling sound continues from beginning to end). Sec. E.5 also details how spatiotemporal priors are injected and shifted over sequential events, further illustrating the well-learned cross-attention mechanism of ST-priors for synchronized video-audio generation.

**JavisDiT supports variant-length sounding video generation.** In the previous sections, our evaluations primarily focused on generating 4-second videos. Here, we extend the assessment to 10-second outputs and report several key metrics. The experimental results in Tab. 5 show that our JavisDiT maintains strong visual and auditory quality (FVD, FAD), semantic consistency (CLIP, CLAP, AVHScore), and audio–video synchrony (JavisScore) when generating 10-second videos. These findings provide strong evidence that our approach effectively supports diverse audio–video generation requirements across different scenarios, durations, and resolutions.

Table 5: **Quantitative evaluation on variant-length generation performance.**

| Length | FVD↓ | FAD↓ | CLIP↑ | CLAP↑ | AVHScore↑ | JavisScore ↑ |
|--------|------|------|-------|-------|-----------|--------------|
| 4s | 241.8 | 7.3 | 0.308 | 0.382 | 0.186 | 0.153 |
| 10s | 233.8 | 7.1 | 0.307 | 0.385 | 0.183 | 0.154 |

## 5.4 HUMAN EVALUATION

In this section, we further conduct user studies to more comprehensively evaluate the performance of joint audio–video generation through human evaluation. Specifically, we randomly sample 100 prompts from JavisBench and run UniVerse-1 (Wang et al., 2025) and our JavisDiT to generate corresponding audio–video outputs. Three volunteers are then recruited to perform blind win–tie–lose preference judgments on three aspects: *video quality*, *audio quality*, and *audio-video alignment*. The averaged scores across annotators are used as the final evaluation metric.

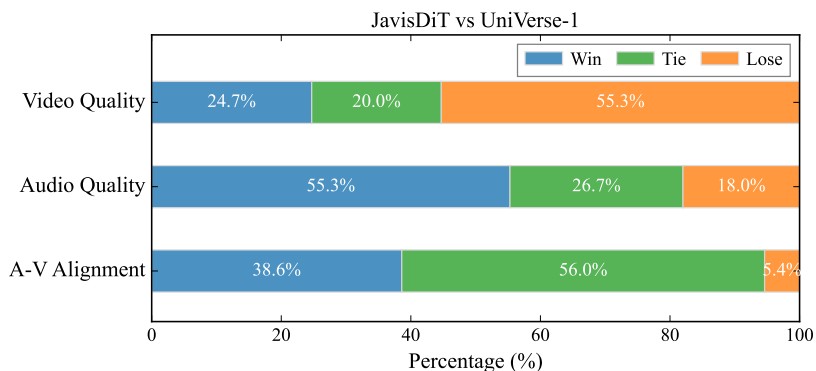

Figure 8: **Human evaluation on generation performance.**

The human evaluation results in Fig. 8 are largely consistent with the objective metrics reported in Tab. 1:

(1) Our video quality is indeed slightly lower than that of UniVerse-1 (Wang et al., 2025). As analyzed in Sec. 5.2, this mainly stems from the performance gap between the backbones: our method currently relies on the OpenSora-1.2 (Zheng et al., 2024) backbone, which is notably weaker than the Wan-2.1 (Wan et al., 2025) backbone used in UniVerse-1. We expect this issue to be substantially alleviated as stronger T2V backbones become available in future versions.

(2) Our audio quality and audio–video alignment outperform UniVerse-1. This is because UniVerse-1 adopts a relatively complex architecture to bridge two separately pretrained branches, and the resulting representation gap weakens cross-modal interaction. In contrast, our JavisDiT design offers a more streamlined and effective mechanism for enabling tight audio–video coupling.

## 6 CONCLUSION

This paper presents JavisDiT, a novel Joint Audio-Video Diffusion Transformer that simultaneously generates high-quality audio and video content with precise synchronization. We introduce the HiST-Sypo Estimator, a fine-grained spatio-temporal alignment module that extracts global and fine-grained priors to guide the synchronization between audio and video. We also propose the JavisBench dataset, comprising 10,140 high-quality text-captioned sounding videos with diverse scenes and real-world complexity, addressing limitations in current benchmarks. In addition, we introduce a temporal-aware semantic alignment mechanism to better evaluate JAVG systems on complex content. Experimental results show that JavisDiT outperforms existing approaches in both content generation and synchronization, establishing a new benchmark for JAVG tasks. Potential limitations and future work are discussed in Sec. A.5.

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

# A    DISCUSSIONS

## A.1    ETHICAL STATEMENT

In this work, we construct our JavisBench from publicly available academic datasets as well as YouTube videos. To ensure ethical compliance, we strictly adhere to YouTube's Terms[1] of Service and licensing policies. Specifically:

1. **Privacy Protection**: We have taken measures to remove any personally identifiable or privacy-sensitive information from the collected data. No private, confidential, or user-specific metadata has been retained.
2. **Copyright Compliance**: All data collection respects the original content licenses. We only utilize publicly accessible videos that are either explicitly licensed for research use or fall under fair-use considerations. No copyrighted content is redistributed or modified in violation of licensing terms.
3. **Responsible Data Release**: Any potential dataset release will fully comply with YouTube's data policies and ethical guidelines. We will **first release the self-curated caption data** to support the evaluation of all metrics except the AV-Quality terms in Tab. 1, and then ensure that shared sounding-videos are either appropriately anonymized or restricted in accordance with relevant regulations and platform policies.

By implementing these measures, we strive to maintain high ethical standards in data collection, use, and dissemination.

## A.2    SOCIETAL IMPACT STATEMENT

This work proposes JavisDiT, a unified model for synchronized audio-video generation from open-ended prompts. While primarily intended as a contribution to foundational multimodal generation research, it has both positive applications and potential societal risks.

**Positive Impact.** JavisDiT may support creative industries by enabling efficient generation of audio-video content. Addressing the challenge of audio-video alignment has practical implications for animation, video conferencing, television broadcasting, and video editing—domains where synchronization is traditionally achieved through offline computation or extensive manual post-processing.

**Potential Risks.** As with other generative models, JavisDiT may be misused to produce highly realistic synthetic media for malicious purposes, including misinformation, impersonation, and manipulation. Its ability to generate temporally coherent sounding videos may increase the realism and persuasiveness of deepfakes. Specifically, fabricated audio-video clips could be used to spread false information or simulate events that never occurred, posing risks to public trust, political discourse, and media integrity.

**Mitigation Strategy.** To mitigate these concerns, we recommend responsible release practices, including usage gating and dataset transparency. Future work may explore watermarking and detection tools to distinguish synthetic output.

## A.3    REPRODUCIBILITY STATEMENT

We provide detailed descriptions of model design, training, and evaluation in both the main paper and the appendix. Furthermore, all code, pretrained checkpoints, and processed datasets will be publicly released to ensure full reproducibility of our results.

## A.4    LLM USAGE STATEMENT

Large Language Models (LLMs) were used solely as writing assistants, including tasks such as language polishing and presentation refinement. They were not involved in the conception of core ideas or designs.

---

[1]https://www.youtube.com/t/terms

Table A1: Latency analysis for 240P, 4s sounding-video generation on H100 with 30 sampling steps.

| Video Branch | Audio Branch | ST-Prior Modulation | AV-Interaction | Overall |
|---|---|---|---|---|
| 13s | 2s | 13s | 2s | 30s |

### A.5 POTENTIAL LIMITATION AND FUTURE WORK

Despite the strong performance in joint audio-video generation, our JavisDiT has several limitations that present opportunities for future research:

1. **Scalability of Training Data**: Our model was trained on 0.6M text-video-audio triplets, which, while substantial, is still limited compared to the scale of some large vision-language models (*e.g.*, OpenSora (Zheng et al., 2024) takes over 60M data to build a foundation mode for video generation). Expanding the dataset with more diverse and higher-quality real-world audio-video samples could further enhance the model's ability to generalize across different domains and fine-grained synchronization patterns.

2. **Synchronization Evaluation Metrics**: While we introduce JavisScore for evaluating audio-video synchronization, its current accuracy of 75% suggests room for improvement. More robust synchronization metrics—potentially incorporating perceptual alignment assessments or human-in-the-loop evaluation—could further refine synchronization quality measurement in JAVG research.

3. **Efficiency and Computation Overhead**: JavisDiT employs a Diffusion Transformer for high-quality generation, but diffusion-based models tend to be computationally intensive, as analyzed in Tab. A1. While our approach achieves state-of-the-art results, generation speed and efficiency remain challenges (*e.g.*, generating a 2-second sounding-video at 720P/24fps/16kHz takes 6 minutes on one H100 GPU). Exploring accelerated sampling strategies or hardware optimization could improve efficiency.

4. **Benchmarking Across Resolutions and Durations**: Our evaluations primarily focus on a fixed resolution (240P) and duration (4s) setting in JavisBench. However, real-world applications may require generation at higher resolutions (*e.g.*, 1080P) and longer durations. Conducting benchmark tests across multiple settings would provide a more comprehensive understanding of current models' strengths and limitations.

By addressing these limitations, future iterations of JavisDiT could further enhance scalability, synchrony, efficiency, and adaptability, pushing the boundaries of joint audio-video generation.

## B RELATED WORK

In the field of AIGC, multimodal generation has become a key topic, encompassing text-to-image (Rombach et al., 2022; Ramesh et al., 2022; Saharia et al., 2022), text-to-video (Ho et al., 2022; Wang et al., 2024d; Yu et al., 2023), and text-to-audio (Liu et al., 2023a; 2024b; Huang et al., 2023) generation, *etc.*. Among these, sounding video generation (Ruan et al., 2023; Xing et al., 2024; Wang et al., 2024b; Sun et al., 2024; Liu et al., 2024a) is drawing increasing attention due to its strong alignment with real-world applications. To generate synchronized audio and video, early works decomposed the task into two cascaded subtasks. a) generating video from text, then adding audio (Comunità et al., 2024; Jeong et al., 2024; Zhang et al., 2024; Ren et al., 2024; Hu et al., 2024); or b) generating audio first, then synchronizing video (Yariv et al., 2024; Jeong et al., 2023; Zhang et al., 2025). As an instance, MovieGen (Polyak et al., 2024) achieves movie-grade generation quality using this cascaded approach. From a methodological perspective, the community focus has shifted from UNet-based models (Rombach et al., 2022; Liu et al., 2024b) to DiT-based methods (Zheng et al., 2024; Polyak et al., 2024) to achieve more state-of-the-art performance.

Another line of work considers the limitations of pipeline approaches, such as error propagation and reliability issues, by focusing on end-to-end Joint Audio-Video Generation (JAVG). This paradigm aims to improve generation quality by modeling audio-video synchronization, including semantic consistency and temporal alignment, and has received growing attention (Ruan et al., 2023; Hayakawa et al., 2024; Ishii et al., 2024). For example, Ruan et al. (2023) introduce a diffusion block that enables cross-modal interaction during denoising. Xing et al. (2024) propose using ImageBind to align the semantic representations of the two modalities. Sun et al. (2024) introduce a hierarchical VAE, mapping audio and video to a shared semantic space while reducing their dimensionality.

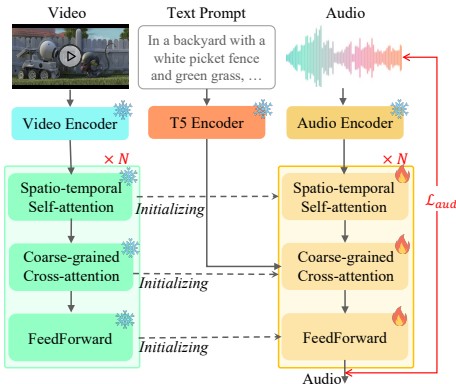

Figure A1: **Audio Pretraining**. Parameters are initialized from the video branch.

Table A2: **Detailed settings for three-stage training.**

| Setting | Stage-1 | Stage-2 | Stage-3 |
|---|---|---|---|
| trainable params | 1.11B | 29.3M | 923.8M |
| learning rate | 1e-4 | 1e-5 | 1e-4 |
| warm-up steps | 1000 | 1000 | 1000 |
| weight decay | 0.0 | 0.0 | 0.0 |
| ema decay | 0.99 | - | 0.99 |
| dropout rate | 0.0 | 0.0 | 0.0 |
| training samples | 788K | 611K | 611K |
| batch size | dynamic | dynamic | dynamic |
| epoch | 55 | 1 | 2 |
| training objective | rect. flow | contrastive | rect. flow |
| GPU days (H100) | 64 | 8 | 256 |

AV-DiT (Wang et al., 2024b) employs a single DiT model to generate both video and audio modalities simultaneously, improving efficiency. Uniform (Zhao et al., 2025) also takes a single DiT for JAVG, by simply concatenating the video and audio latent tokens during the diffusion process, without any explicit synchronization guidance. SyncFlow (Liu et al., 2024a) utilizes STDiT (Zheng et al., 2024) blocks to enhance the video generation quality, with a temporal adapter to guide the audio generation process. Unfortunately, current JAVG methods either lack a strong backbone for audio and video generation or insufficiently model audio-video synchronization, resulting in suboptimal generation quality. To address these issues, we construct a novel DiT-based JAVG model, further enhanced by modeling hierarchical spatial-temporal synchronized prior features. Moreover, this paper aims to further advance JAVG by providing a more comprehensive, robust, and challenging benchmark.

## C  IMPLEMENTATION DETAILS

### C.1  JAVISDIT MODEL CONFIGURATION

**Model Architecture.** As Fig. 2 illustrates, JavisDiT consists of two branches for video and audio generation, each comprising $N = 28$ DiT blocks. Within each DiT block, the latent video and audio representations are processed through several modules, where all attention modules utilize 16 attention heads with a hidden size of 1152. The intermediate dimension of the FFN is $4\times$ the hidden size. Each attention and FFN module is preceded by a LayerNorm layer. In practical implementation, every attention module is followed by an FFN module to further enhance feature modeling. The video/audio latent is sequentially processed with: *Spatial-SelfAttn – CrossAttn – Spatial-CrossAttn – Bi-CorssAttn – FFN – Temporal-SelfAttn – CrossAttn – Temporal-CrossAttn – Bi-CorssAttn – FFN* for $N = 28$ times, resulting in our JavisDiT with 3.14B parameters in total.

In particular, we adopt spatial–temporal (self- or cross-) attention modules primarily to reduce computational cost. For video tokens of size $(T \times H \times W)$, the computational complexity of full self-attention is $O\big((THW)^2\big)$. In contrast, spatial–temporal attention reduces the complexity to $O\big(T(HW)^2 + T^2(HW)\big) = O\big(THW \cdot (T + HW)\big)$, which substantially lowers both memory consumption and computational overhead.

**Training Strategy.** As Sec. 3.3 states, we carefully design a three-stage training strategy to achieve high-quality single-modal generation and ensure fine-grained spatio-temporal synchrony on generated videos and audios. In particular, the video branch of JavisDiT (including the ST-SelfAttn module, the Coarse-Graind CrossAttn module, and the FFN module) is initialized from OpenSora (Zheng et al., 2024) and frozen during the whole training stages. We use the weights of the video branch to initialize the audio branch for better convergence (see Fig. A1). The detailed configuration for the three-stage training strategy is displayed in Tab. A2.

**Training Data Curation.** For audio-pretraining (stage-1), the audio-caption pairs come from various public datasets, including AudioSet (Gemmeke et al., 2017b), AudioCaps (Kim et al., 2019), VGGSound (Chen et al., 2020a), WavCaps (Mei et al., 2024), Clotho (Drossos et al., 2020), ESC50 (Piczak, 2015), GTZAN (Sturm, 2013), MACS (Martín-Morató & Mesaros, 2021), and

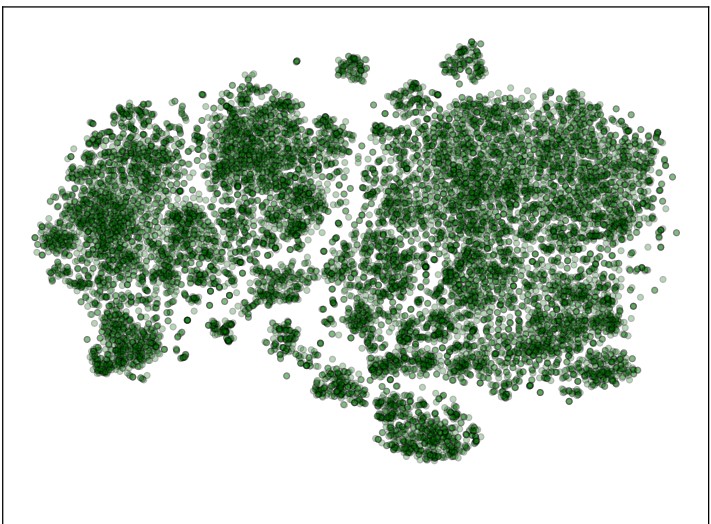

Figure A2: **T-SNE visualization of prompt distribution of the training dataset.**

UrbanSound8K (Salamon et al., 2014). They contribute to a total of 788K training entries. For the ST-Prior estimator and the final JavisDiT training (stage-2&3), the data source comes from two newly-proposed sounding video datasets: MMTrail (Chi et al., 2024) and TAVGBench (Mao et al., 2024). MMTrail provides slightly more caption annotations (2M) than TAVGBench (1.7M) but has lower quality. Due to the resource limit, we collect a part of the original sounding-videos from YouTube and filtered out around 80% human-talking videos with the FunASR [2] tool. We finally collect 136K high-resolution video-audio-caption triplets from MMTrail and 475K from TAVGBench, resulting in 611K data for training in total. Fig. A2 visualizes the prompt distribution, which further demonstrates the diversity of our collected training dataset. The data construction for ST-Prior estimator's contrastive learning is discussed in Sec. C.2.4.

**Training Objective and Inference.** The ST-Prior Estimator is trained with contrastive learning objectives, which will be detailed in Sec. C.2. For the DiT model, we use rectified flow (Liu et al., 2023b) as the denoising scheduler for better performance (Zheng et al., 2024; Polyak et al., 2024). The inference sample step is 30, with classifier-free guidance at 7.0. The video and audio latent are concurrently sampled at each inference step. The video encoder-decoder comes from OpenSora (Zheng et al., 2024), and the audio encoder-decoder comes from AudioLDM2 (Liu et al., 2024b). Both of them are frozen during training.

**Detailed Evaluation Setup.** In Sec. 5 in our manuscript, we conduct a comprehensive evaluation across multiple video and/or audio generation models on two settings: (1) our proposed Javis-Bench benchmark and (2) previously used AIST++ (Li et al., 2021) and Landscape (Lee et al., 2022) datasets. For our JavisBench, generative models are required to generate videos and 240p and 24fps, with audios at 16kHz sample rate. The duration is 4 seconds. For AIST++ (Li et al., 2021), we follow the literature (Ruan et al., 2023; Sun et al., 2024) to generate 2-second clips at the visual resolution of $256 \times 256$, with 240p on landscape (Lee et al., 2022).

## C.2 ST-PRIOR ESTIMATOR CONFIGURATION

### C.2.1 OVERVIEW

Since ImageBind (Girdhar et al., 2023) has built a consistent semantic space across multiple modalities, we use ImageBind's text encoder to extract the potential spatio-temporal prior from the input text caption. Formally, for a given input text $s$, we leverage the 77 hidden states from the last layer of ImageBind's text encoder, and learn $N_s$ spatial tokens $\mathbf{p}_s$ and $N_t$ temporal tokens $\mathbf{p}_t$ from a 4-layer transformer block $\mathcal{P}$. Notably, since the same event may occur at different locations and timestamps in different video-audio generations, the same text $s$ can also produce different ST-Priors

---

[2]https://github.com/modelscope/FunASR

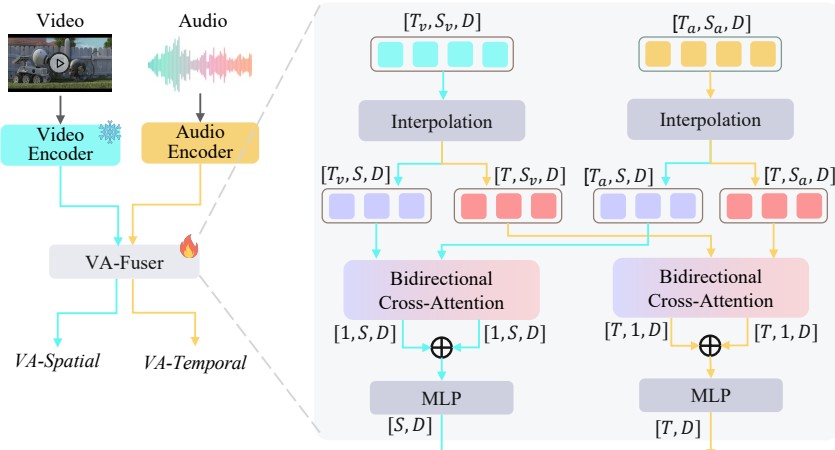

Figure A3: **Architecture of VA-Fuser to encode spatio-temporal embeddings.**

$(\mathbf{p}_s, \mathbf{p}_t)$. We adopt a sampling strategy akin to VAE, where our ST-Prior Estimator $\mathcal{P}$ outputs the mean and variance of a Gaussian distribution, from which we sample a plausible $(\mathbf{p}_s, \mathbf{p}_t)$ to describe specific spatio-temporal events. This can be formalized as: $(\mathbf{p}_s, \mathbf{p}_t) = \mathcal{P}_\phi(s; \epsilon)$, where $\epsilon$ refers to the normalized Gaussian distribution.

As Fig. 3 states, the ST-Prior Estimator $\mathcal{P}_\phi$ is trained with contrastive learning. Specifically, we take the text prompts $s$ and estimated prior $(\mathbf{p}_s, \mathbf{p}_t)$ as an *anchor*, and treat the corresponding synchronous video-audio $(\mathbf{v}, \mathbf{a})$ pairs from training datasets as *positive* samples. Then, we either synthesize an asynchronous video $\mathbf{v}^-$ or audio $\mathbf{a}^-$, and take the original $\mathbf{a}$ or $\mathbf{v}$ to form *negative* pairs.

Formally, we also extract $N_s/N_t$ positive or negative spatial/temporal embeddings from synchronous or asynchronous video-audio pairs: $(\mathbf{e}_s^+, \mathbf{e}_t^+) = \mathcal{E}_\psi(\mathbf{v}, \mathbf{a})$, $(\mathbf{e}_s^-, \mathbf{e}_t^-) = \mathcal{E}_\psi(\mathbf{v}, \mathbf{a}^-)$ or $\mathcal{E}_\psi(\mathbf{v}^-, \mathbf{a})$. Sec. C.2.2 displays more details.

The training objective is to make the priors $(\mathbf{p}_s, \mathbf{p}_t)$ closer to the synchronous embeddings $(\mathbf{e}_s^+, \mathbf{e}_t^+)$ while pushing them further from the asynchronous $(\mathbf{e}_s^-, \mathbf{e}_t^-)$, thereby equipping the priors $(\mathbf{p}_s, \mathbf{p}_t)$ with fine-grained conditioning capabilities. Several contrastive losses with the Kullback-Leibler (KL) divergence loss are utilized to optimize the spatial and temporal priors (see Sec. C.2.3):

$$\mathcal{L}_{prior} = \mathcal{L}_{\text{contrast}}(\mathbf{p}_s, \mathbf{e}_s^+, \mathbf{e}_s^-) + \mathcal{L}_{\text{kl}}(\mathbf{p}_s) + \mathcal{L}_{\text{contrast}}(\mathbf{p}_t, \mathbf{e}_t^+, \mathbf{e}_t^-) + \mathcal{L}_{\text{kl}}(\mathbf{p}_t). \tag{A1}$$

The details for negative sample construction are provided in Sec. C.2.4.

### C.2.2    SPATIAL-TEMPORAL VA-ENCODING

For a given video clip with the corresponding audio, we first use OpenSora's (Zheng et al., 2024) and AudioLDM2's (Liu et al., 2024b) VAE-encoders to respectively map the raw data to the feature space, and derive a lightweight *VA-Fuser* module to extract the spatio-temporal embeddings from the video-audio pair. As shown in Fig. A3, the raw video $V \in \mathbb{R}^{T \times (H \times W) \times C}$ will be encoded and reshaped to a video embedding $\mathbf{v} = E_v(V) \in \mathbb{R}^{T_v \times S_v \times D_v}$, where the downsampling rate for $(T, H, W)$ is $(4, 8, 8)$, and the output channel $D = 4$. Similarly, the raw audio will be first transformed to the MelSpectrogram $A \in \mathbb{R}^{R \times M \times 1}$, and then encoded to an audio embedding $\mathbf{a} = E_a(A) \in \mathbb{R}^{T_a \times S_a \times D_a}$. We use a linear layer to unify video and audio channels into the hidden size $D = 512$.

Then, we align the video and audio embeddings with the pre-defined spatial-temporal token numbers via linear interpolation. Specifically, we will obtain a video-spatial embedding $\mathbf{v}_s \in \mathbb{R}^{T_v \times S \times D}$, a video-temporal embedding $\mathbf{v}_t \in \mathbb{R}^{T \times S_v \times D}$, a audio-spatial embedding $\mathbf{a}_s \in \mathbb{R}^{T_a \times S \times D}$, and a audio-temporal embedding $\mathbf{a}_t \in \mathbb{R}^{T \times S_a \times D}$. Subsequently, we apply bidirectional attention (see Fig. 2(d)) for the spatial va-embedding-pair $(\mathbf{v}_s, \mathbf{a}_s)$ and the temporal va-embedding-pair $(\mathbf{v}_t, \mathbf{a}_t)$. After averaging the temporal dimension $(T_v, T_a)$ and merging the video-audio embeddings, we use a 2-layer MLP module to project to the desired video-audio *spatial embedding* $\mathbf{e}_s \in \mathbb{R}^{S \times D}$. With similar operations, we can obtain the video-audio *temporal embedding* $\mathbf{e}_t \in \mathbb{R}^{T \times D}$.

As defined in Sec. 2, the extracted *spatial embedding* $\mathbf{e}_s$ should determine the occurrence of a specific event in a region of the video frame, with the corresponding appearance of matching frequency components in the audio spectrogram. On the other hand, the extracted *temporal embedding* $\mathbf{e}_t$ should identify the onset or termination of an event at a specific frame or timestamp in the video, with the corresponding start or stop of the response in the audio. We achieve these goals by a dual contrastive learning objective.

### C.2.3 CONTRASTIVE TRAINING LOSSES

Our core objective is to align the spatial/temporal priors $(\mathbf{p}_s, \mathbf{p}_t)$ (extracted from text) more closely with the positive spatial/temporal embeddings $(\mathbf{e}_s^+, \mathbf{e}_t^+)$ derived from synchronized video-audio pairs, while pushing them away from the embeddings $(\mathbf{e}_s^-, \mathbf{e}_t^-)$ of negative samples (asynchronous video-audio pairs). This ensures that the priors $(\mathbf{p}_s, \mathbf{p}_t)$ provide the fine-grained spatio-temporal condition required for synchronized video-audio generation.

To achieve this, we designed four contrastive loss functions. For simplicity, we omit the subscripts of $s, t$ in this part, as we apply the same loss functions on both spatial and temporal priors/embeddings.

1. *Token-level hinge loss*: $\mathcal{L}_{token}(\mathbf{p}, \mathbf{e}^+, \mathbf{e}^-) = |1.0 - sim(\mathbf{p}, \mathbf{e}^+)| + |1.0 + sim(\mathbf{p}, \mathbf{e}^-)|$, where $sim$ refers to the cosine similarity.
2. *Auxiliary discriminative loss*: $\mathcal{L}_{disc}(\mathbf{p}, \mathbf{e}^+, \mathbf{e}^-) = \mathcal{L}_{BCE}(\mathcal{D}_\theta(\mathbf{p}, \mathbf{e}^+), 1) + \mathcal{L}_{BCE}(\mathcal{D}_\theta(\mathbf{p}, \mathbf{e}^+), 0)$, where $\mathcal{D}_\theta$ is a 1-layer attention module parameterized with $\theta$ that use the $[CLS]$ token to gather the information in $\mathbf{p}$ and $\mathbf{e}$.
3. *VA-embedding discrepancy loss*: $\mathcal{L}_{vad}(\mathbf{e}^+, \mathbf{e}^-) = |1.0 + sim(\mathbf{e}^+, \mathbf{e}^-)|$, which aims to enlarge the discrepancy of positive va-embedding $\mathbf{e}^+$ and negative $\mathbf{e}^-$.
4. *L2-regularization loss*: $\mathcal{L}_{reg}(\mathbf{p}, \mathbf{e}^+) = \|\mathbf{p} - \mathbf{e}^+\|_2$, which directly pushes the prior $\mathbf{p}$ towards the positive (synchronous) va-embedding $\mathbf{e}^+$.

Overall, the contrastive learning loss is combined with:

$$\mathcal{L}_{\text{contrast}}(\mathbf{p}, \mathbf{e}^+, \mathbf{e}^-) = \mathcal{L}_{token}(\mathbf{p}, \mathbf{e}^+, \mathbf{e}^-) + \mathcal{L}_{disc}(\mathbf{p}, \mathbf{e}^+, \mathbf{e}^-) + \mathcal{L}_{vad}(\mathbf{e}^+, \mathbf{e}^-) + \mathcal{L}_{reg}(\mathbf{p}, \mathbf{e}^+). \tag{A2}$$

Specifically, we take the text prompts $s$ and estimated prior $(\mathbf{p}_s, \mathbf{p}_t)$ as an *anchor*, and treat the corresponding synchronous video-audio $(\mathbf{v}, \mathbf{a})$ pairs from training datasets as *positive* samples. Then, we either synthesize an asynchronous video $\mathbf{v}^-$ or audio $\mathbf{a}^-$, and take the original $\mathbf{a}$ or $\mathbf{v}$ to form *negative* pairs. It can be formulated as $(\mathbf{e}_s^+, \mathbf{e}_t^+) = \mathcal{E}_\psi(\mathbf{v}, \mathbf{a})$, $(\mathbf{e}_s^-, \mathbf{e}_t^-) = \mathcal{E}_\psi(\mathbf{v}, \mathbf{a}^-)$ or $\mathcal{E}_\psi(\mathbf{v}^-, \mathbf{a})$. The VA-Encoder $\mathcal{E}_\psi$ is detailed in Sec. C.2.2, and the negative sample construction is introduced in Sec. C.2.4. We also discuss the efficacy of each loss function in Sec. E.4

### C.2.4 NEGATIVE SAMPLE CONSTRUCTION

Given a synchronized video-audio pair from the training set, we design two approaches to synthesize *easy* and *hard* asynchronous videos/audios for optimization. (1) For *easy* negatives, we take AudioLDM2 (Liu et al., 2024b) to generate arbitrary audios from the text without referencing the corresponding video, which naturally results in asynchronous video-audio pairs. (2) For *hard* negatives, various augmentation strategies are employed to add, remove, or modify elements in either the original video or audio from GT-pairs, creating more fine-grained spatio-temporally asynchronous pairs. As detailed below, these methods further enhance the discriminative power of the ST-Prior, and Fig. A4 and Fig. A5 showcase some representative augmentation examples.

**Video Spatial Augmentation**

- *Random Masking* (Fig. A4(A)). Videos are divided into $6 \times 6$ grids, with a mask ratio $p$ uniformly sampled from $(0.2, 0.8)$). A proportion $p$ of the grids is randomly masked.
- *Adding Subject Trajectories* (Fig. A4(B)). The trajectories from the SA-V dataset (Ravi et al., 2024) are overlaid to simulate new sound-producing objects. Trajectories are preprocessed using RepViT-SAM (Wang et al., 2024a) for interpolation from 6fps to 24fps, retaining objects with an average size $> 32 \times 32$ pixels. During augmentation, a random trajectory is selected from the pool and added to the video.

**Video Temporal Augmentation**

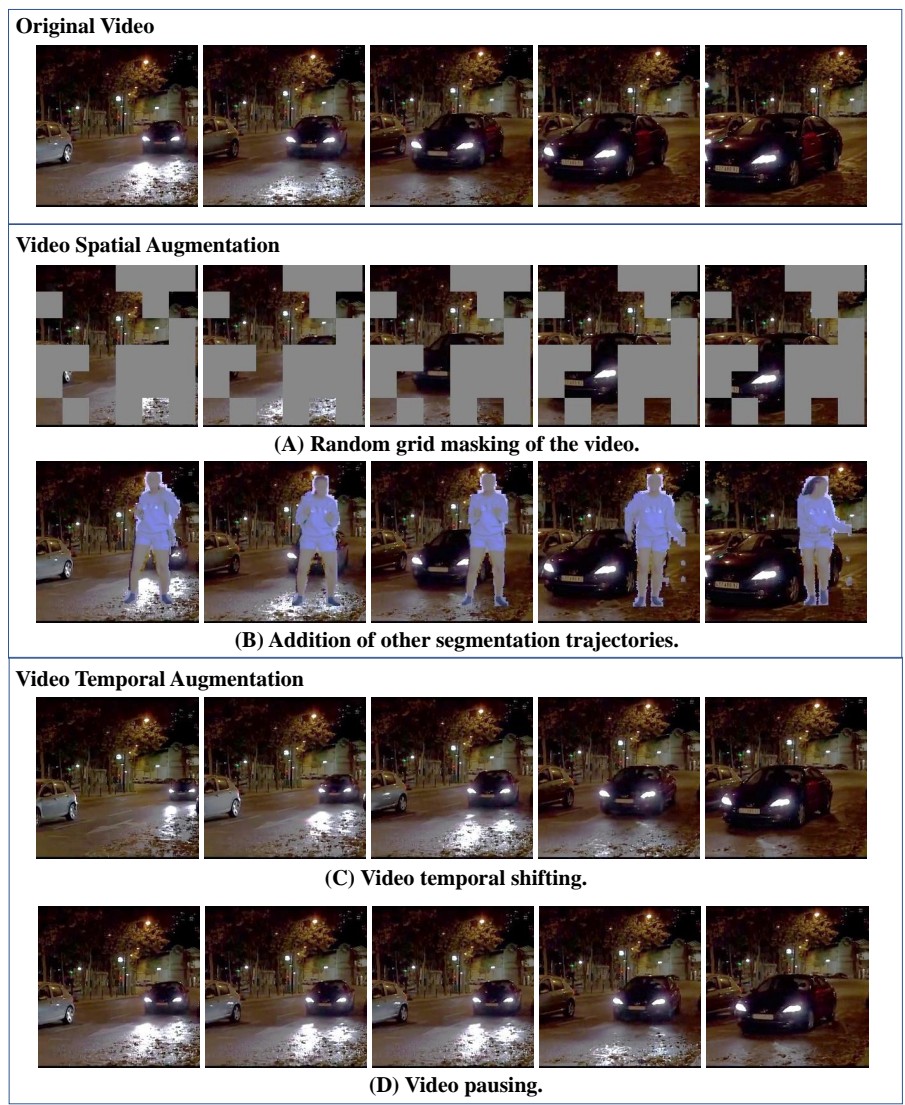

Figure A4: **Video augmentation for spatial and temporal negative samples.**

- *Video Temporal Shifting* (Fig. A4(C)). The video is cyclically rearranged from a random starting point to create temporal misalignment.
- *Video Pausing* (Fig. A4(D)). A random frame is duplicated for at least 0.5 seconds at a chosen point, pausing the video and disrupting temporal synchronization.

**Audio Spatial Augmentation**

All audio augmentations (except temporal shifting) focus on sound-producing intervals, avoiding silent or noisy segments. We use QwenPlusAPI[3] to extract potential sound-producing objects from audio descriptions, and utilize AudioSep (Liu et al., 2024c) to separate sound sources and labels intervals using RMS. Separation results are precomputed and stored as metadata, guiding target augmentations.

- *Audio Source Addition* (Fig. A5(A)). Additional audio from other dataset sources is mixed with the original one, causing spatial content misalignment between audio and video.
- *Audio Source Removal* (Fig. A5(B)). A separated sound source from the original audio is randomly removed to induce more spatial misalignments.

---

[3]https://help.aliyun.com/zh/model-studio/developer-reference/use-qwen-by-calling-api

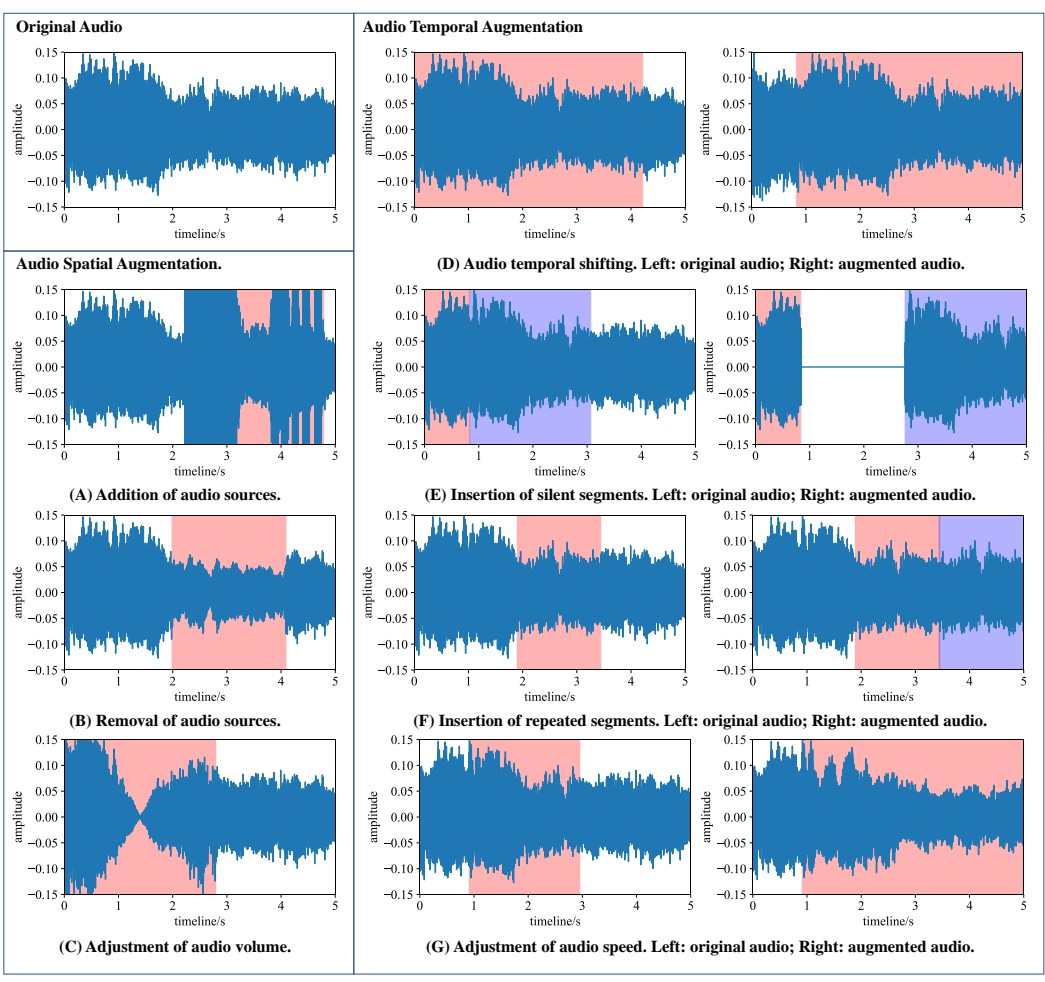

Figure A5: **Audio augmentation for spatial and temporal negative samples.**

- *Audio Volume Adjustment* (Fig. A5(C)). The target audio's volume is modified by a randomly selected transformation (*e.g.*, cosine, sine, linear increase/decrease) to adjust its amplitude.

**Audio Temporal Augmentation**

- *Audio Temporal Shifting* (Fig. A5(D)). Audio is shifted by extracting a new interval from the original, creating temporal misalignment.
- *Silent Segment Insertion* (Fig. A5(E)). Silent segments are inserted into the target interval, disrupting synchronization with the video timeline in subsequent frames.
- *Repeated Segment Insertion* (Fig. A5(F)). Target segments are repeated after a random transformation interval, making another type of timeline misalignment.
- *Audio Speed Adjustment* (Fig. A5(G)). Playback speed within the target interval is altered to $0.5\times$ or $2\times$, causing another type of timeline desynchronization.

# D    MORE DETAILS ON JAVISBENCH

## D.1    MOTIVATION

A strong generative model must produce diverse video content and audio types while ensuring fine-grained spatio-temporal synchronization. However, we have noticed that current benchmarks and metrics lack sufficient diversity and robustness for comprehensive evaluation. For instance, the commonly used AIST++ (Li et al., 2021) and Landscape (Lee et al., 2022) benchmarks focus

only on limited scenarios (human dancing for AIST++ and natural scenarios for landscape) with merely 20-100 test samples. On the other hand, although TAVGBench (Mao et al., 2024) provides 3,000 test samples (but not released yet), it lacks detailed analysis of different scenarios in practical audio-video generations. While larger-scale datasets such as VGGSound (Chen et al., 2020a) and AudioSet (Gemmeke et al., 2017a) provide millions of video-audio pairs for training and testing, they are typically limited to a single-event taxonomy (*e.g.*, human voice, engine) and do not reflect multi-event dynamics within a single video clip. Consequently, they are not suitable for evaluating spatial-temporal alignment in complex real-world environments. Moreover, all videos in VGGSound are captured in the wild, resulting in a lack of in-house scenes. Both of the two datasets fall short in scene diversity, underrepresenting diverse categories such as 2D/3D animations and industrial settings, which are crucial for evaluating generative models under varied and challenging conditions. In addition, the video-audio synchronization metric AV-Align (Yariv et al., 2024) cannot produce accurate evaluation on complicated scenarios, *e.g.*, with multiple sounding events (Ishii et al., 2024; Mao et al., 2024).

We are therefore motivated to propose a more challenging benchmark to evaluate generative models from various dimensions on a larger data scale in Sec. 4.1, with a particular focus on spatio-temporal synchronization. We also introduce a robust metric to quantify the fine-grained spatio-temporal alignment between generated video and audio pairs in Sec. 4.2. Here we provide more details of the benchmark construction and metric verification.

## D.2 TAXONOMY

To comprehensively evaluate the capabilities of joint video-audio generation models, we designed five evaluation dimensions from different aspects: (1) *Event Scenario*, (2) *Video Style*, (3) *Sound Type*, (4) *Spatial Subjects*, and (5) *Temporal Composition*. Leveraging GPT-4 (Achiam et al., 2023), we enumerated 3-5 primary categories under each aspect, contributing to a hierarchical categorization system with 5 dimensions and 19 categories. Tab. A3 presents the detailed definitions and clarifications.

## D.3 DATA CURATION

### D.3.1 COLLECTION

We collected data from two sources to construct the benchmark. First, we incorporate existing datasets' test sets, including those in the JAVG domain like Landscape (Lee et al., 2022) and AIST++ (Li et al., 2021) (despite their limited coverage) and others from sounding-video understanding tasks such as FAVDBench (Shen et al., 2023) and AVSBench (Zhou et al., 2022). Second, we expanded the benchmark by crawling videos uploaded to YouTube between June 2024 and December 2024 to prevent data leakage in previous methods' training scope (Mao et al., 2024). Using the previously defined categorization system, we prompt GPT4 to generate potential keywords for specific categories, enabling targeted video collection and avoiding indiscriminate gathering of homogeneous data, which significantly improves curation efficiency. This stage yields around 30K sounding video candidates.

### D.3.2 QUALITY-BASED PRE-FILTERING

Following OpenSora (Zheng et al., 2024), we utilize a series of filtering tools to improve data quality, including:

1. *Scene Cutting*. Given the extended duration of some YouTube videos (up to several hours), we employed PySceneDetect[4] for scene detection and segmented the videos at identified scene transitions. Each clip was constrained to a length of 2–60 seconds, resulting in approximately 230K sounding video clips.
2. *Aesthetic Filtering*. We filter out videos with aesthetic scores (Schuhmann et al., 2021) lower than 4.5, remaining in 70K clips.
3. *Optical-flow Filtering*. We use UniMatch (Xu et al., 2023) to estimate the motion quality of the given videos, and remove the (static) videos whose score is lower than 0.1. This produces 46K candidates.

---

[4]https://github.com/Breakthrough/PySceneDetect

Table A3: **Clarification of the category taxonomy of our JavisBench**.

| Aspect | Category | Description and Examples |
|---|---|---|
| **Event Scenario** | Natural Scenario | Scenes dominated by natural environments with minimal human interference, such as forests, oceans, and mountains. |
| | Urban Scenario | Outdoor spaces shaped by human activity, including cities, villages, streets, and parks. |
| | Living Scenario | Indoor environments where daily human activities occur, like houses, schools, and shopping malls. |
| | Industrial Scenario | Work-oriented spaces related to industrial or energy activities, such as factories, construction sites, and mines. |
| | Virtual Scenario | Imaginative or abstract settings, including virtual worlds, sci-fi cities, and artistic installations. |
| **Visual Style** | Camera Shooting | Filmed with handheld, fixed, or drone cameras, including slow-motion footage. |
| | 2D-Animate | Styles like hand-drawn animation, flat animation, cartoon styles, or watercolor illustrations. |
| | 3D-Animate | Photorealistic styles, sci-fi/magical effects, CG (Computer Graphics), or steampunk aesthetics. |
| **Sound Type** | Ambient Sounds | Sounds that occur naturally in the environment, including both natural and human-influenced surroundings. This category includes sounds like wind, rain, water flow, animal sounds, human activity (*e.g.*, traffic, construction), and urban noise. |
| | Biological Sounds | Sounds produced by living creatures (*e.g.*animals, birds). This includes vocalizations such as barking, chirping, growling, as well as non-vocal human sounds like heartbeat, and other physical noises. |
| | Mechanical Sounds | Sounds generated by man-made machines, devices, or mechanical processes. This includes the noise of engines, motors, appliances, and any mechanical or electronic noise. This category also includes malfunction sounds (*e.g.*, malfunctioning machinery or alarms). |
| | Musical Sounds | Sounds related to music or musical performance, including both human-generated and instrument-generated sounds and melodies. This category covers singing, instrumental performances, as well as background music used in various media formats. |
| | Speech Sounds | Sounds generated from human speech, whether in conversation, dialogue, public speeches, debates, interviews, or monologues. This category specifically covers linguistic communication in various contexts, whether formal, informal, or contentious. |
| **Spatial Composition** | Single Subject | There is only one primary object or source producing sound in the scene. |
| | Multiple Subject | There are multiple primary objects that (or potentially can) make sounds in the scene. |
| | Off-screen Sound | The source of the sound is not visible in the scene but logically exists (*e.g.*, a car engine outside the camera view). |
| **Temporal Composition** | Single Event | The audio contains only one event, with no overlapping sounds. For example, "a single dog barking without background noise." |
| | Sequential Events | There are multiple events occurring sequentially, with no overlap. For example, "the applause begins after the music performance ends." |
| | Simultaneous Events | Multiple audio sources are present simultaneously, such as "a person speaking while music plays in the background." |

4. *OCR Filtering*. We use DBNet (Liao et al., 2022) to detect and filter out the videos containing more than 5 text regions, resulting in a smaller 30K scope.

5. *Speech Filtering*. As the Internet-available videos contain too much human speech (including talking and voiceover), we use FunASR[5] to detect and remove the speech videos. In this step, there are still around 20% speech videos remaining, due to the non-perfect speech detection by FunASR. We keep this part of videos in our final benchmark to ensure the diversity of video-audio sources. This step leads to 22.3K sounding video clips.

### D.3.3 ANNOTATION

Since most of the collected data lacks captions, we designed a generic pipeline to generate captions for the video-audio pairs, and then categorize them into corresponding classes from different aspects, as illustrated in Sec. D.2.

---

[5]https://github.com/modelscope/FunASR

Table A4: **Data source composition before and after filtering strategies of our JavisBench**

|  | YouTube | FAVDBench | AVSBench | Landscape | AIST++ | Total |
|---|---|---|---|---|---|---|
| **Before Filtering** | 30,107 | 1,000 | 804 | 100 | 20 | 32,031 |
| **After Filtering** | 8,507 | 833 | 680 | 100 | 20 | 10,140 |

- First, we use Qwen2-VL-72B (Wang et al., 2024c) to generate detailed captions for videos. Metadata from the data source (*e.g.*, object labels or YouTube keywords) is included in the context to enhance caption quality.
- Next, we use Qwen2-Audio-7B (Chu et al., 2024) to caption the audio in detail. We do not incorporate the previously generated video captions as contextual input, as it produces even more hallucinations with the input bias (*i.e.*, wrongly identifying a sound that will happen in corresponding visual scenarios but actually does not exist in the current audio).
- Then, we employ Qwen2.5-72B-Instruct (Yang et al., 2024a) to merge video and audio captions into a unified text prompt, serving as the textual condition for the JAVG task. During this process, Qwen2.5-72B is prompted to reason on the video-audio captions and identify apparent logical mistakes (*e.g.*, there is a dog in the video but Qwen2-Audio gives sounds from a car engine) or missing captions (*e.g.*, unknown sound source).
- Finally, we query Qwen2.5-72B-Instruct again to classify the data points based on the video, audio, and generated captions, assigning each entry to the appropriate category in the classification system. We do not prompt multimodal LLMs to do this categorization because the generative models to evaluate will only receive the text caption as inputs.

After removing the logical conflict captions and classification results that fail to parse, we obtain 19.4K sounding video clips with detailed captions and hierarchical categorization results.

### D.3.4 CONTENT-BASED POST-FILTERING

To build a diversified and balanced benchmark to evaluate joint audio-video generation, we conduct another post-filtering based on the categorization results obtained above. In particular, we further remove videos that only contain background music and speech voice. By doing so, nearly half of the videos are filtered out, since a large part of YouTube videos rely on music and voice acting to attract viewers. After human checking, we obtained 10,140 samples in our JavisBench with diverse data sources and fine-grained category annotations, setting a new standard to facilitate a comprehensive evaluation for future joint audio-video generation (JAVG) methods. The category statistics can be found in Fig. 4, and Tab. A4 displays the data sources before and after our filtering strategies.

### D.4 EXTENDED SPECIFICATION ON JAVISSCORE EVALUATION SUITE

### D.4.1 IMPLEMENTATION DETAILS

Given a generated video-audio pair $(V, A)$, we use ImageBind (Girdhar et al., 2023) to estimate audio-visual synchrony with following steps:

1. **Temporal Segmentation via Sliding Windows**. $(V, A)$ is chunked into several segments with 2-seconds window size and 1.5-seconds overlap ($\mathcal{C} = \{(V_1, A_1), (V_2, A_2), \cdots (V_W, A_W)\}$). The 2-second window ensures compatibility with the audio encoder's minimum processing length (Girdhar et al., 2023), preventing suboptimal feature extraction. The 1.5-second overlap enhances continuity and robustness, allowing each frame to be evaluated within multiple temporal contexts. This mitigates artifacts caused by abrupt segmentation boundaries.
2. **Frame-wise Audio-Visual Similarity Computation**. Inspired by Mao et al. (2024), we calculate the cosine similarity between each frames ($\mathcal{F} = \{V_{i,1}, V_{i,2}, \cdots V_{i,w}\}$) and the whole audio clip $A_i$ by using ImageBind's vision and audio encoders ($E_v, E_a$). Frame-wise similarity captures fine-grained temporal dynamics, ensuring transient events (*e.g.*, lip movements, object interactions) are accurately modeled.
3. **Segment-wise Synchronization Estimation**. Instead of averaging all similarity scores, we select the 40% least synchronized frames (with lower similarities) and compute their mean to obtain the synchronization score for each window. By focusing on the least synchronized

     frames, the metric becomes more sensitive to local resynchronization, as a simple mean can be biased by a considerable proportion (*e.g.*, 70%) of synchronized frames.

4. **Global Synchronization Estimation**. The final JavisScore is computed by averaging the window-level scores across all segments, balancing local variations while maintaining sensitivity to desynchronization patterns. Due to window overlap, each video frame is evaluated multiple times, reducing the influence of outliers and providing a more stable final score.

These steps can be formulated as:

$$S_{Javis} = \frac{1}{W} \sum_{i=1}^{W} \sigma(V_i, A_i), \quad \sigma(V_i, A_i) = \frac{1}{k} \sum_{j=1}^{k} \underset{\min}{\text{top-}k} \{\cos\left(E_v(V_{i,j}), E_a(A_i)\right)\}. \quad (A3)$$

This approach effectively differentiates synchronized and desynchronized video-audio pairs.

### D.4.2 VERIFICATION ON METRIC QUALITY

This paper provides a quantitative comparison between the accuracy of video-audio synchrony metrics. We constructed a validation dataset consisting of 3,000 audio-video pairs to evaluate the effectiveness of our proposed metric, JavisScore. The validation set is initialized by 1,000 positive (synchronous) video-audio pairs from our JavisBench dataset, and we construct around 2,000 negative pairs from three different sources:

1. The first 1,000 negative pairs are generated through online augmentation/transformation on the positive pairs. Given a synchronized video-audio pair $(V^+, A^+)$, we reuse the augmentation strategies in Sec. C.2.4 to generate asynchronous video-audio pairs, *i.e.*, $(V^+, A^-)$ or $(V^-, A^+)$. The four augmentation types (video-spatial, video-temporal, audio-spatial, audio-temporal) produce 250 negative samples, resulting in the first 1,000 negative pairs.

2. The second 500 negative pairs come from separate generations. In particular, we randomly split the 1,000 positive samples into two parts, and select the first 500 samples to prompt AudioLDM2 (Liu et al., 2024b) to generate audios conditioned solely on text captions, without accessing the corresponding videos. This naturally leads to 500 asynchronous video-audio pairs.

3. The third 500 negative video-audio pairs are generated by a preliminary JAVG model conditioned on the other 500 text prompts from positive samples. These generated pairs were then manually annotated to determine whether they were synchronized or not. We finally obtained 411 hard negative samples with 89 positive video-audio pairs.

In the evaluation dataset, each positive sample is related to 2 negative video-audio pairs. One is constructed by online augmentation on the positive entry (the first negative source), and the other is generated from text captions of the positive data point (the second or third negative source).

Then, we compare our proposed JavisScore with the previously-used AV-Align (Yariv et al., 2024) metric on the 3,000 evaluation samples, and the result is displayed in Tab. A5. We calculate the AV-Align scores and our JavisScore on all the 3000 samples, and compute the AUROC value for the binary classification (ideally, a VA-synchrony metric should produce higher scores on positive video-audio pairs), which shows JavisScore achieves approximately 0.13 higher AUROC than AV-Align. Then, as each positive sample is related to 2 negative video-audio pairs,

Table A5: Comparison of VA-synchrony metrics. † means we relax the criteria to $Acc = \mathbb{I}\{s^+ \geq s^-\}$.

| Metric | AUROC | Accuracy |
|---|---|---|
| *Random-Guess* | *0.5000* | *0.5000* |
| AV-Align | 0.5296 | 0.5254 |
| DeSync | 0.5742 | 0.3961 |
| DeSync † | 0.5742 | 0.7797 |
| JavisScore | 0.6533 | 0.7514 |

we can calculate the prediction accuracy of $1000 \times 2 = 2000$ paired positive-negative samples. It is viewed as correct when a metric assigns a higher score to the positive sample than to the corresponding negative video-audio pair. According to Tab. A5, our metric significantly surpasses AV-Align by 23% accuracy, demonstrating the efficacy of video-audio synchrony measurement.

Notably, AV-Align nearly performs as a random guess (with a near 0.5 AUROC and Accuracy). This is because AV-Align simply uses optical flow to capture the video dynamics, and match with audio dynamics estimated by onset detection results. If there is a concurrent "pulse" at the video and audio timelines, AV-Align will produce a high synchrony score. However, in real-world scenarios, the video

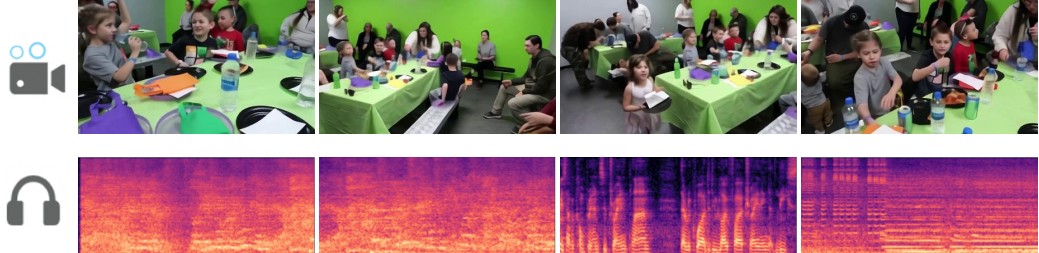

Figure A6: **Complex Scenario with multi-source sounds at a time.**

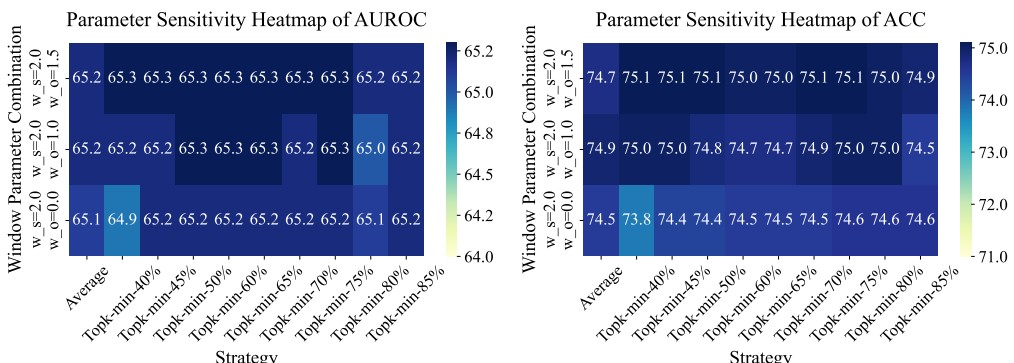

Figure A7: **Parameter sensitivity evaluation of our JavisScore.** Our method presents stable and robust video-audio synchrony estimate at various settings. We finally choose (2-second window size, 1.5-second overlap, topmin-40%) due to the relatively better performance.

optical flow and audio onset detection cannot precisely capture the start and end of the visual-audio event. As exemplified in Fig. A6, when a person in the video speaks, the movements of their mouth are subtle and often too minimal to be effectively captured by optical flow. Moreover, in environments with strong background noise (*e.g.*, TV programs), the onset of human talking also becomes difficult to detect. The scores generated by AV-Align are unreliable in such complex scenarios. In contrast, our proposed JavisScore leverages the high-dimensional semantic space of ImageBind (Girdhar et al., 2023) to compute synchronization at the second-level, robustly distinguishing between synchronized and unsynchronized cases. It is worth noting, however, that our metric does not achieve 100% accuracy. Developing a more precise evaluation metric remains a significant challenge in the JAVG domain, and we hope this work will inspire further advancements in this critical area.

In addition, SynchFormer (Iashin et al., 2024) employs a 21-category classifier to estimate the temporal shift degree between video and audio, which is then used to compute a synchrony score (lower is better). However, as shown in Tab. A5, its discrete predictions result in poor separation between positive and negative samples, yielding low accuracy and AUROC. Only when we relax the evaluation metric to a $Acc = \mathbb{I}\{s^+ \geq s^-\}$ criterion does its accuracy approach that of JavisScore. Besides, SynchFormer is inherently limited to predicting temporal shifts and is not applicable to the diverse spatiotemporal misalignment scenarios considered in the real world (see Sec. C.2.4). Nevertheless, its approach suggests a promising direction: learning a neural network–based synchrony metric via supervision, which we leave as future work.

Besides, we also conducted an ablation study to evaluate the parameter sensitivity of our JavisScore, including the sliding window size, overlap length, and score computation strategy. As suggested by Fig. A7, the optimal parameters were identified as a sliding window size of 2, an overlap length of 1.5, and selecting the top 40% minimum strategy. However, other settings do not significantly reduce the performance of our metric. The worst accuracy, for example, still achieves around 74% and outperforms AV-Align of 52.5% by a large margin. The results further demonstrate the robustness of our JavisScore metric.

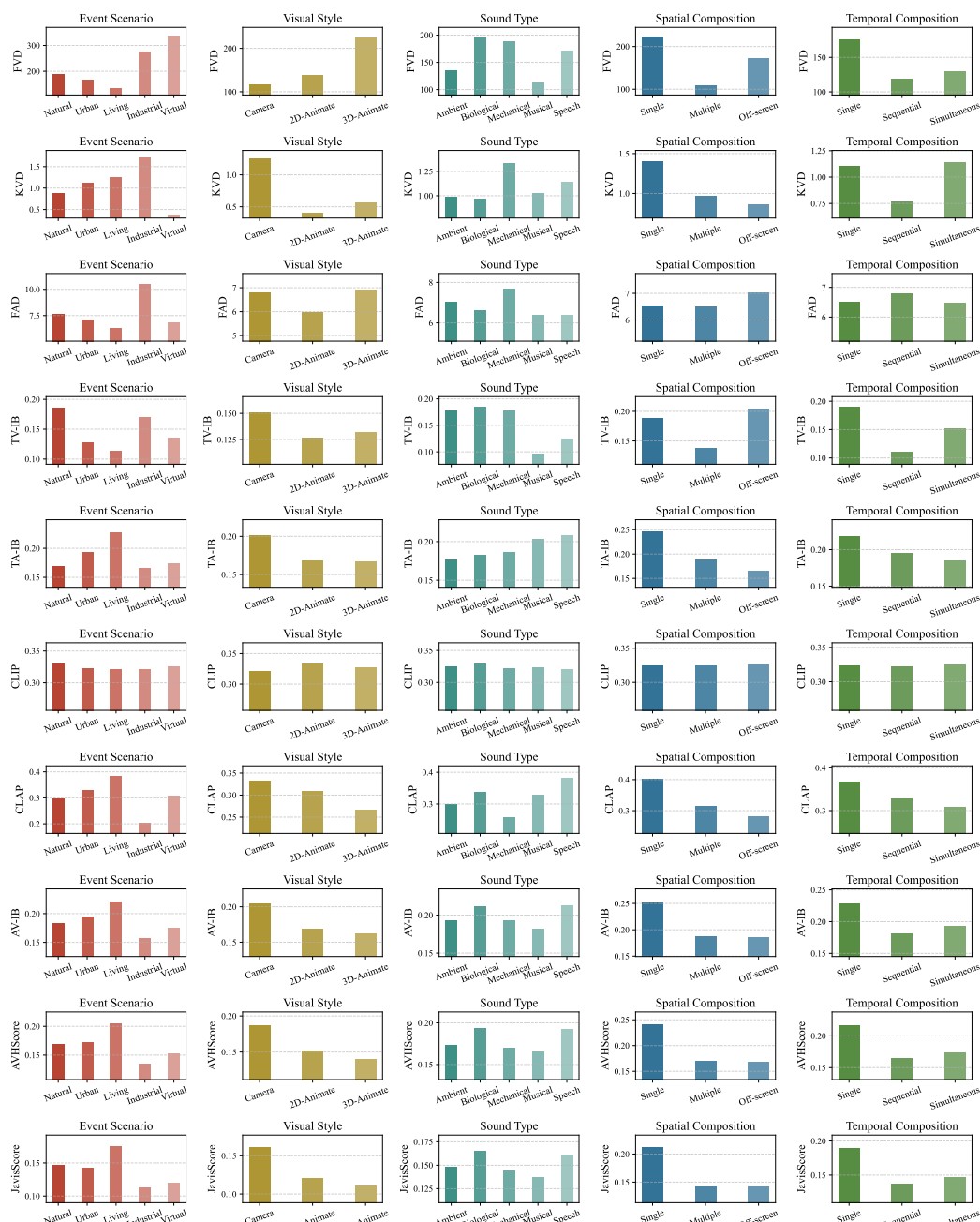

Figure A8: **Metric distribution on all JavisBench's taxonomy**.

## D.5  IN-DEPTH ANALYSIS OF EVALUATION RESULTS

Fig. A8 provides a more fine-grained, multi-dimensional, and comprehensive evaluation of our model on JavisBench, offering an in-depth analysis of the current SOTA models' limitations in real-world joint audio-video generation tasks. Based on these results, we summarize the following key insights:

- **Insufficient unimodal modeling capability in rare scenarios**: (1) The FVD in Event Scenario (1st row, 1st column) indicates poor video quality in *industrial* and *virtual* scenes, suggesting weaker modeling capability in these domains. (2) The FVD in Visual Style (1st row, 2nd column) shows a significant disparity between generated and real videos in *3D-animate* scenes, potentially due to insufficient training data. (3) The FAD in Sound Type (r3c3) suggests weaker audio

Table A6: **Detailed evaluation results of ablation on model design (Tab. 3.)**

| STDiT | HiST-Sypo | BiCA | AV-Quality | | | Text-Consistency | | | | AV-Consistency | | | AV-Synchrony |
|---|---|---|---|---|---|---|---|---|---|---|---|---|---|
| | | | FVD↓ | KVD↓ | FAD↓ | TV-IB↑ | TA-IB↑ | CLIP↑ | CLAP↑ | AV-IB↑ | CAVP↑ | AVHScore↑ | JavisScore↑ |
| × | × | × | 442.0 | 3.9 | 10.5 | 0.268 | 0.139 | 0.287 | 0.344 | 0.197 | 0.796 | 0.147 | 0.118 |
| √ | × | × | 335.4 | 3.4 | 8.1 | 0.276 | 0.145 | **0.309** | 0.380 | 0.200 | 0.799 | 0.156 | 0.130 |
| √ | √ | × | 249.7 | **2.9** | 7.4 | 0.275 | **0.147** | 0.305 | 0.375 | 0.207 | 0.800 | 0.184 | 0.150 |
| √ | × | √ | 269.1 | 3.1 | 7.9 | 0.273 | 0.144 | 0.300 | 0.377 | 0.200 | 0.795 | 0.162 | 0.133 |
| √ | √ | √ | **241.8** | **2.9** | **7.3** | **0.277** | 0.146 | 0.308 | **0.382** | **0.209** | **0.801** | **0.186** | **0.153** |

Table A7: **Detailed evaluation results of ablation on ST-Prior design (Tab. 4.)**

| $N_s$ | $N_t$ | Injection | AV-Quality | | | Text-Consistency | | | | AV-Consistency | | | AV-Synchrony |
|---|---|---|---|---|---|---|---|---|---|---|---|---|---|
| | | | FVD↓ | KVD↓ | FAD↓ | TV-IB↑ | TA-IB↑ | CLIP↑ | CLAP↑ | AV-IB↑ | CAVP↑ | AVHScore↑ | JavisScore↑ |
| 0 | 0 | - | 269.1 | 3.1 | 7.9 | 0.273 | 0.144 | 0.300 | 0.377 | 0.200 | 0.795 | 0.162 | 0.133 |
| 1 | 1 | CrossAttn | 289.9 | 3.2 | 8.1 | 0.269 | 0.140 | 0.293 | 0.375 | 0.204 | 0.797 | 0.175 | 0.137 |
| 16 | 16 | CrossAttn | 257.2 | 3.0 | 7.5 | 0.275 | 0.144 | 0.305 | 0.381 | **0.209** | **0.801** | 0.185 | 0.151 |
| 32 | 32 | CrossAttn | **241.8** | **2.9** | 7.3 | **0.277** | **0.146** | 0.308 | **0.382** | **0.209** | **0.801** | **0.186** | **0.153** |
| 32 | 32 | Addition | 252.7 | 3.0 | 7.4 | 0.274 | 0.142 | 0.303 | 0.380 | 0.206 | 0.798 | 0.180 | 0.144 |
| 32 | 32 | Modulate | 255.0 | 2.9 | 7.4 | **0.277** | 0.143 | **0.309** | **0.382** | 0.207 | 0.800 | 0.180 | 0.145 |

modeling in *ambient*, *biological*, and *mechanical* categories, likely because these categories are too broad. This observation aligns with Event Scenario (r3c1), where *natural* and *industrial* scenes also exhibit high FAD values.

- **Unimodal quality does not directly correlate with text consistency**: (1) The TV-IB in Visual Style (r4c2) reveals that *2D/3D-animate* scenes exhibit poor text-following capability, despite their unimodal quality. (2) The CLAPScore in Sound Type (r7c3) suggests weak audio-text alignment in *ambient* and *mechanical* scenes, potentially due to a lack of corresponding audio-text pairs in the first-stage (Audio Branch) training data.
- **Poor AV-synchronization in challenging scenarios**: (1) The JavisScore in Event Scenario, Visual Style, and Sound Type (r10c1-3) show that scenarios with poor unimodal quality also suffer from weak audio-video synchronization, particularly in *virtual environments*, *2D/3D-animate styles*, and *musical sound types*. (2) The JavisScore in Spatial and Temporal Composition (r10c4-5) indicate significantly lower AV synchronization performance for *complex events* (*e.g.*, sequential/simultaneous events, multiple sound sources, or off-screen sounds) compared to *simpler scenes* (*e.g.*, single events with a single sound source).

In conclusion, current SOTA models still struggle with *rare and complex scenarios*, both in terms of *audio-video generation quality* and *synchronization performance*. The JAVG community still faces significant challenges in bridging the gap between research models and real-world applications.

# E   ADDITIONAL EXPERIMENTS

## E.1   FULL EVALUATION RESULTS FOR ABLATION STUDIES

In the manuscript, we use several proxy metrics ($S_{AVQ}$; $S_{AVC}$; $S_{AVS}$) to simplify the presentation when evaluating the quality, consistency, and synchrony of generated audio–video outputs. In this section, we provide the full results corresponding to Tab. 3 (model-design ablations) and Tab. 4 (ST-Prior ablations) in Tab. A6 and Tab. A7, respectively, to further enhance the clarity and completeness of the paper.

## E.2   EVALUATION ON AUDIO GENERATION QUALITY

Tab. A8 presents the evaluation of our model's performance in the first training stage, concerning both unimodal text-to-audio generation quality and text-following capability. Specifically, we employ two datasets for evaluation:

- **AudioCaps test set** (Kim et al., 2019): Filtered by AudioLDM2 (Liu et al., 2024b), this dataset consists of 964 samples, each 8–10 seconds long. Models are required to generate 10-second and 4-second audio clips.

Table A8: **Evaluation on audio generation.** After sufficient training iterations at stage 1, our JavisDiT presents moderate audio generation performance on in-domain test set AudioCaps (Kim et al., 2019) and comparable quality on out-of-domain test set JavisBench-mini.

| Method | AudioCaps (InD, 10s) | | | AudioCaps (InD, 4s) | | | JavisBench-mini (OoD, 4s) | | |
|---|---|---|---|---|---|---|---|---|---|
| | FAD↓ | TA-IB↑ | CLAP↑ | FAD↓ | TA-IB↑ | CLAP↑ | FAD↓ | TA-IB↑ | CLAP↑ |
| AudioLDM2 | *2.01* | *0.205* | *0.487* | 5.57 | 0.147 | 0.326 | 8.81 | **0.153** | 0.360 |
| JavisDiT-audio-ep13 | 5.88 | 0.145 | 0.319 | - | - | - | 8.33 | 0.150 | 0.368 |
| JavisDiT-audio-ep55 | 5.19 | 0.164 | 0.356 | 6.23 | 0.141 | 0.301 | **8.11** | 0.152 | **0.381** |

- **JavisBench-mini**: A subset containing 1,000 samples from JavisBench ranging from 2 to 10 seconds in length, which is also utilized in ablation studies in Sec. 5.3). Models are required to generate 4-second audio clips.

On these datasets, we first analyze the performance progression of our model across different training stages and then compare it against the current state-of-the-art (SOTA) model, AudioLDM2 (Liu et al., 2024b).

As shown in Tab. A8, with increased training iterations (from epoch 13 to epoch 55), our model demonstrates improvements in both *audio generation quality* (*e.g.*, FAD decreases from 5.88 to 5.19 on AudioCaps) and *text consistency* (*e.g.*, TA-IB increases from 0.145 to 0.164, while CLAPScore improves from 0.368 to 0.381 on JavisBench-mini). Furthermore, our model achieves comparable audio generation performance to AudioLDM2 on JavisBench-mini, further validating the effectiveness of our DiT-based architecture.

Notably, our model performs worse than AudioLDM2 on AudioCaps, which can be attributed to two primary factors: (1) *Training strategy*: AudioLDM2 is explicitly trained for 10-second audio generation, whereas our model supports variable-length audio generation. In particular, although one can forcibly alter AudioLDM2's generation length during inference by directly modifying the shape of the noisy latent (*e.g.*, changing it to 4 seconds), the model's denoising process becomes misaligned with its training distribution, leading to a significant drop in generation quality, as shown in Tab. A8. In contrast, since our audio pretraining is performed under variable-length settings, the generation quality at 4 seconds remains much closer to that of AudioLDM2. We believe that collecting more data and training for additional iterations will further improve generation performance across different lengths. (2) *Potential overfitting problem*: The training data of AudioLDM2 includes the training set of AudioCaps, leading to a smaller in-domain test gap, which increases the risk of overfitting. In contrast, JavisBench is collected from more diverse real-world YouTube audio sources, and its domain-specific soundscapes are not exposed to either AudioLDM2 or our model during training. Thus, performance on the *out-of-domain* JavisBench-mini provides a more realistic reflection of a model's usability in real-world scenarios.

In future iterations, we will continue enhancing the unimodal audio generation quality, as this is a crucial prerequisite for achieving precise *video-audio synchronization*.

### E.3 INVESTIGATION OF ST-PRIOR'S EXTRACTION AND REPRESENTATION

In Fig. A9, we investigate the representation capability of ST-priors from three aspects.

Firstly, since the latent representations of video ($v$) and audio ($a$) contain different numbers of spatial and temporal tokens (*e.g.*, for a 240P, 24fps, 16kHz, 4-second sounding video, $v \in \mathbb{R}^{400 \times 30 \times C_v}$ while $a \in \mathbb{R}^{16 \times 64 \times C_a}$), we also experimented with various token allocation strategies for the spatial and temporal priors. These included *ratios* such as 1:2 (n16x32), 1:1 (n32x32), and 2:1 (n32x16). As shown in Fig. A9, the 1:1 ratio achieved the best performance. This could be attributed to the inherent disparity in the latent shapes: video latent contains significantly more spatial tokens than temporal tokens, while audio latent stays in the opposite situation. Thus, a balanced 1:1 prior token ratio helps minimize bias.

Next, using the 1:1 ratio, we explored different configurations of prior *numbers* and *dimensions*. The results in Fig. A9 indicate that increasing the prior number and dimension consistently improve

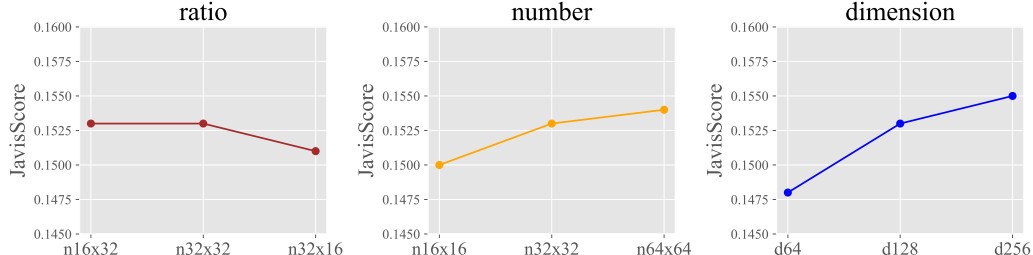

Figure A9: **Further ablation on ST-Prior hyper-parameters,** including (1) spatial-temporal token ratio, (2) token number, and (3) embedding dimension. Our default setting of "n32x32-d128" is a good trade-off between performance and training cost.

Table A9: **Ablation of ST-Prior loss functions.** All training objectives jointly contribute to video-audio synchronization.

| $\mathcal{L}_{token}$ | $\mathcal{L}_{disc}$ | $\mathcal{L}_{vad}$ | $\mathcal{L}_{reg}$ | AV-Consistency | | | AV-Synchrony | |
|---|---|---|---|---|---|---|---|---|
| | | | | IB-AV↑ | CAVP↑ | AVHScore↑ | AV-Align↑ | JavisScore↑ |
| √ | × | × | × | 0.190 | 0.799 | 0.167 | 0.097 | 0.133 |
| √ | √ | × | × | 0.193 | 0.799 | 0.170 | 0.102 | 0.136 |
| √ | √ | √ | × | 0.196 | 0.800 | 0.174 | 0.096 | 0.140 |
| √ | √ | × | √ | 0.202 | 0.800 | 0.179 | 0.107 | 0.149 |
| √ | √ | √ | √ | **0.209** | **0.801** | **0.186** | **0.122** | **0.153** |

performance, demonstrating the scalability of our approach. In this work, we chose the "n32x32+d128" configuration as it offers a good trade-off between performance and training cost, considering the diminishing returns of further scaling.

### E.4    TRAINING LOSSES FOR ST-PRIOR'S ESTIMATION

In Sec. C.2.3, we introduce four loss functions to train our ST-Prior Estimator: (1) *Token-level hinge loss*: $\mathcal{L}_{token}$, (2) *Auxiliary discriminative loss*: $\mathcal{L}_{disc}$, (3) *VA-embedding discrepancy loss*: $\mathcal{L}_{vad}$, and (4) *L2-regularization loss*: $\mathcal{L}_{reg}$. This section presents a detailed ablation study on the efficacy of utilized loss functions.

According to Tab. A9, $\mathcal{L}_{disc}$ brings slight improvement in addition to the original $\mathcal{L}_{token}$ (*e.g.*, 0.193 *vs.* 0.190 for IB-AV), as they share the same optimization purpose —— pushing the text prior anchor to the positive video-audio samples while pushing away from negative samples —— and differ only in the specific gradient back-propagation mechanism.

On the other hand, $\mathcal{L}_{vad}$ considerably enhances the synchrony of generated video-audio pairs (*e.g.*, 0.140 *vs.* 0.136 on JavisScore), thanks to its ability to maximize the divergence between positive and negative video-audio samples themselves. However, $\mathcal{L}_{reg}$ offers even greater benefits due to its smoother regularization, facilitating the convergence of the text prior to the positive embeddings.

Moreover, the combination of all loss functions achieves the best performance (*e.g.*, 0.153 of JavisScore), as they collaboratively address the same goal from different perspectives: embedding video-audio synchrony into the text prior and ensuring it captures the semantic meaning of synchronization. This comprehensively validates both our motivation and the effectiveness of our methodology.

### E.5    ADDITIONAL VISUALIZATION OF ST-PRIOR INJECTION

In Fig. A10, we include a more detailed generation example to illustrate how attention shifts during sequential ST-Prior injection:

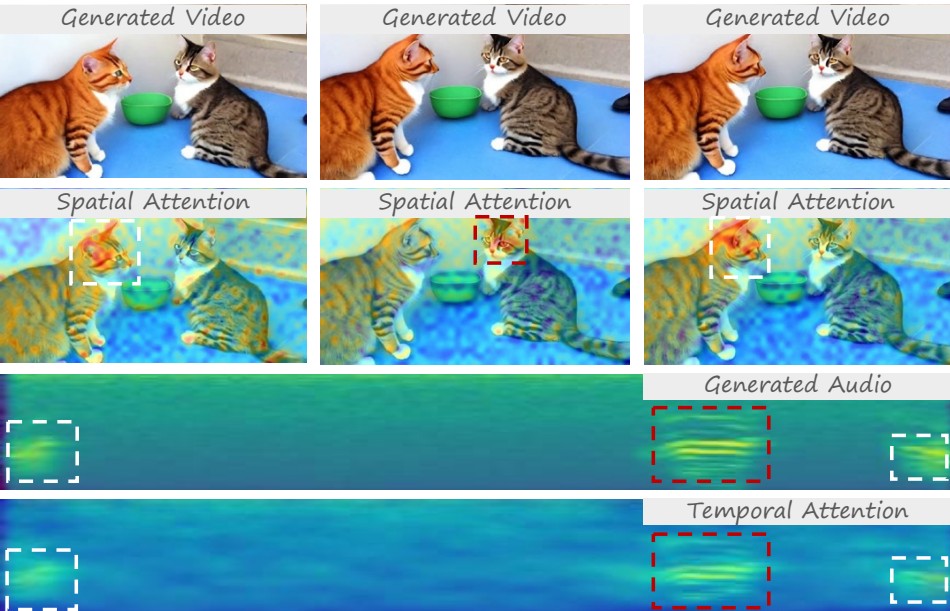

Figure A10: Attention shifts over sequential audiovisual events during ST-Prior injection.

- First, when the orange cat on the left meows, the visual spatial attention concentrates on its head, which corresponds to the region highlighted by the white dashed box in the audio-attention map.
- Then, when the tabby cat on the right meows (indicated by the red dashed box in the audio spectrogram), the visual attention shifts to the tabby's head, providing a more direct spatial prior.
- Finally, when the orange cat meows again, both spatial and temporal attention shift once more, injecting updated prior information into the audio and video branches.

This example more clearly demonstrates how spatiotemporal priors are injected and shifted over sequential events, further validating the effectiveness of our method.

### E.6 MORE GENERATION EXAMPLES

Fig. A11 presents realistic audio-visual pairs generated by our JavisDiT across diverse scenarios, including industrial, outdoor, and natural environments. These results encompass a wide range of visual styles, such as camera-captured footage, 2D/3D animations, as well as various audio types, including mechanical, musical, ambient, and biological sounds. Our model effectively maintains video-audio synchronization across cases involving single or multiple sounding subjects, as well as single, sequential, or simultaneous sounding events. All multimedia resources are available in the supplementary materials.

### E.7 DISCUSSION: EXTENSION TO X-CONDITIONAL GENERATION

Built on diffusion models with transformers (DiT), our JavisDiT can be easily extended to support various conditional video-audio generation tasks, as shown in Fig. A12. Inspired by UL2 (Tay et al., 2022) and OpenSora (Zheng et al., 2024), we propose a dynamic masking strategy to support video and audio conditioning, from the basic (I) *text-to-audio-video* (t2av) generation to (II) *audio-to-video* (a2v) generation, (III) *video-to-audio* (v2a) generation, (IV) *audio-image-to-video* (ai2v) generation, (V) *image-to-audio-video* (i2av) generation, and (VI) *audio-video-extension* (av_ext) generation. Note that the text condition still works in all kinds of conditional generations, which provides both coarse-grained global semantic embeddings and fine-grained spatio-temporal priors on targeted visual-sounding events. In particular, we unmask the specific video and frames to be conditioned on for X-conditional generation. During both model training and inference, unmasked frames will have timestep 0, while others remain the same (t):

- Unmasking all video/audio frames for v2a/a2v generation.
- Unmasking the first frame of the video (and all audio frames) for i2av (and ia2v) generation.

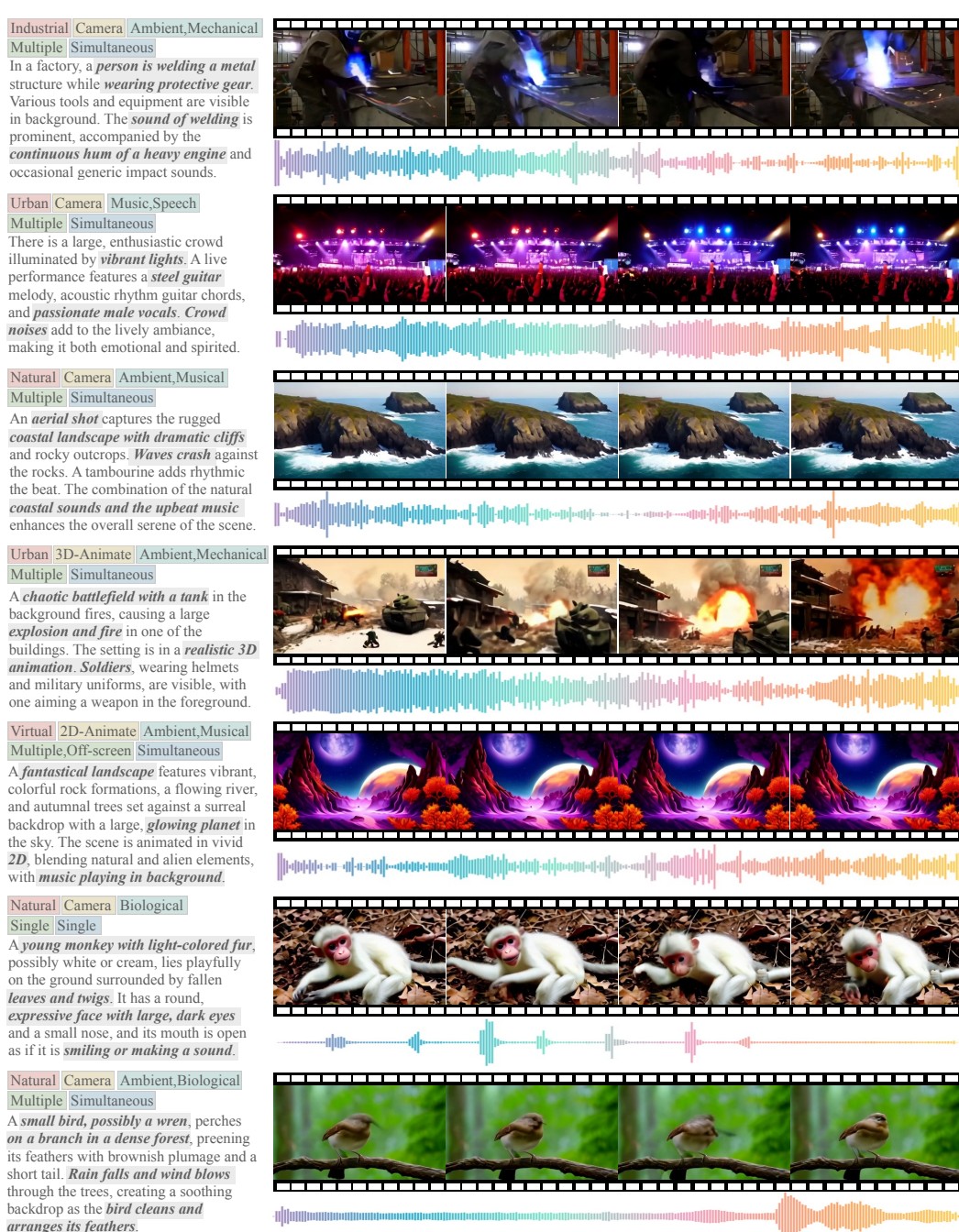

**Industrial** | Camera | Ambient,Mechanical | Multiple | Simultaneous
In a factory, a ***person is welding a metal*** structure while ***wearing protective gear***. Various tools and equipment are visible in background. The ***sound of welding*** is prominent, accompanied by the ***continuous hum of a heavy engine*** and occasional generic impact sounds.

**Urban** | Camera | Music,Speech | Multiple | Simultaneous
There is a large, enthusiastic crowd illuminated by ***vibrant lights***. A live performance features a ***steel guitar*** melody, acoustic rhythm guitar chords, and ***passionate male vocals***. ***Crowd noises*** add to the lively ambiance, making it both emotional and spirited.

**Natural** | Camera | Ambient,Musical | Multiple | Simultaneous
An ***aerial shot*** captures the rugged ***coastal landscape with dramatic cliffs*** and rocky outcrops. ***Waves crash*** against the rocks. A tambourine adds rhythmic the beat. The combination of the natural ***coastal sounds and the upbeat music*** enhances the overall serene of the scene.

**Urban** | 3D-Animate | Ambient,Mechanical | Multiple | Simultaneous
A ***chaotic battlefield with a tank*** in the background fires, causing a large ***explosion and fire*** in one of the buildings. The setting is in a ***realistic 3D animation***. ***Soldiers***, wearing helmets and military uniforms, are visible, with one aiming a weapon in the foreground.

**Virtual** | 2D-Animate | Ambient,Musical | Multiple,Off-screen | Simultaneous
A ***fantastical landscape*** features vibrant, colorful rock formations, a flowing river, and autumnal trees set against a surreal backdrop with a large, ***glowing planet*** in the sky. The scene is animated in vivid ***2D***, blending natural and alien elements, with ***music playing in background***.

**Natural** | Camera | Biological | Single | Single
A ***young monkey with light-colored fur***, possibly white or cream, lies playfully on the ground surrounded by fallen ***leaves and twigs***. It has a round, ***expressive face with large, dark eyes*** and a small nose, and its mouth is open as if it is ***smiling or making a sound***.

**Natural** | Camera | Ambient,Biological | Multiple | Simultaneous
A ***small bird, possibly a wren***, perches ***on a branch in a dense forest***, preening its feathers with brownish plumage and a short tail. ***Rain falls and wind blows*** through the trees, creating a soothing backdrop as the ***bird cleans and arranges its feathers***.

Figure A11: **Extensive JAVG cases on diverse event scenarios, visual styles, audio types, sounding subjects, and temporal compositions.** Our JavisDiT achieves high-quality and text-consistency for single-modality generation and keeps a good video-audio synchronization.

- Unmasking a preceding part of video and audio frames for va-extension.

We view the integration of all audio-video interactive generation tasks in one unified model as our future work.

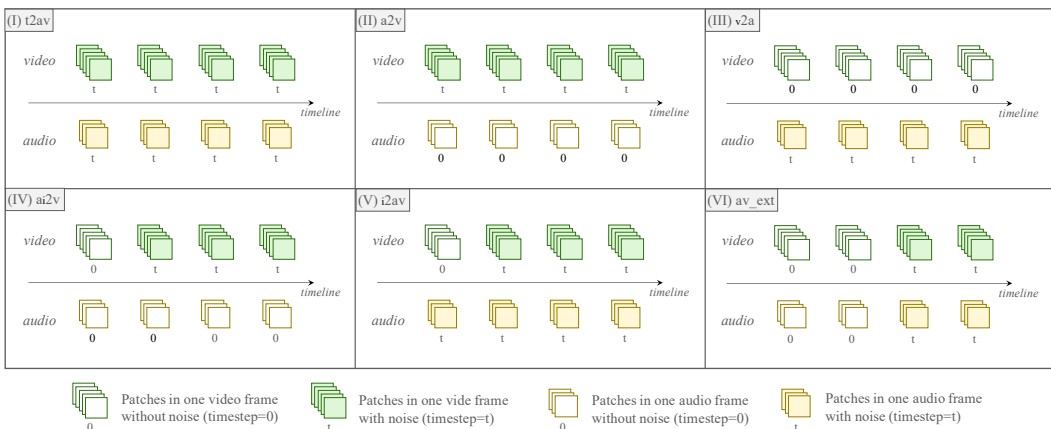

Figure A12: **Masking strategies for X-conditional generation.** The DiT architecture allows feasible conditional video-audio generation by replacing the noisy latent representation with reference videos and/or audios with specific strategies during training and inference.

