# OpenReview forum: "JavisDiT: Joint Audio-Video Diffusion Transformer with Hierarchical Spatio-Temporal Prior Synchronization"
_ICLR.cc/2026/Conference — ICLR 2026 Poster_

### Official Review · Reviewer_AE6y · 2025-10-26

**Soundness:** 3
**Presentation:** 3
**Contribution:** 3
**Rating:** 8
**Confidence:** 4

**Summary:**

This paper presents JavisDiT, an end-to-end audio-video synchronized generation model based on the Diffusion Transformer architecture. By introducing a Hierarchical Spatial-Temporal Synchronized Prior (HiST-Sypo), the model achieves fine-grained alignment between visual and auditory signals, significantly improving spatial-temporal consistency and overall generation quality. The authors also construct a large-scale dataset, JavisBench, and propose a new synchronization evaluation metric, JavisScore, which enhances the systematicity and objectivity of the study. Experimental results show that JavisDiT outperforms existing methods across multiple benchmarks. Overall, this work is solid with new innovations and extensive experiments, demonstrating potential application in multimodal generation.

**Strengths:**

- The research topic in this paper is meaningful and valuable. Audio-video joint generation (JAVG) is a promising emerging field, and an open-sourced JAVG method is appreciated.
- The proposed model exhibits some novel ideas. In particular, the proposed Hierarchical Spatial-Temporal Synchronized Prior (HiST-Sypo) enables fine-grained modeling of spatial-temporal synchronization relationships in generation, which is a new and promising attempt.
- The data and evaluation contributions are also great. The proposed JavisBench benchmark and JavisScore metric may provide valuable resources and evaluation standards for future research in the community.

**Weaknesses:**

Given the clear motivation and solid contributions in method, data, and benchmark, I have no major concerns. I would like to share some questions or suggestions for further improvement. Specifically,

- The model structure may be complex and resource-intensive. The dual-branch Diffusion Transformer architecture for audio and video involves a large number of parameters, leading to potentially high training and inference costs. The proposed HiST-Sypo requires additional pre-training. It is recommended to include a complexity and resource consumption analysis in both training and inference.

- The paper lacks failure case analysis. Although the proposed method is effective, it would be better to include typical failure cases and corresponding discussions, which would help reveal the model’s limitations and potential directions for improvement.

- There is a slight overclaim in some parts of the writing. The current work mainly focuses on sound effects generation and has not yet been extended to speech audio. It is suggested to avoid overextended descriptions of timbre or speech-like attributes in the Introduction section to maintain precision and academic rigor.

- Minor issues: 1) The appendix contains comprehensive details. But, it may be better to move the ablation table 4 into the appendix while adding more details back to the main paper (eg, to further explain or validate the key design of HiST-Sypo).  2) Typo: '240P4s' in the Caption of Table A1 at Line 810 should be '240P 4s'?

**Questions:**

- An open question is whether it would be possible to use much less data to achieve comparable performance?

- Regarding the text semantic understanding module: In related works such as Tango [1] (text-to-audio model) and MultiTalk [2] (speech-based audio-visual generation model), the text encoder is often based on T5-like large language models [3,4], which can capture richer semantic representations. Has the author considered adopting similar LLM architectures to further enhance text semantic modeling capability?

[1]Text-to-audio generation using instruction guided latent diffusion model. ACM MM.

[2]  Let Them Talk: Audio-Driven Multi-Person Conversational Video Generation. arXiv preprint arXiv:2505.22647.

[3] Scaling instruction-finetuned language models. Journal of Machine Learning Research, 25(70), pp.1-53.

[4] UniMax: Fairer and More Effective Language Sampling for Large-Scale Multilingual Pretraining. ICLR.

---

> ### Author Response · Authors · 2025-11-26
> **Response to Reviewer AE6y**
>
> We thank the reviewer for the valuable time and effort to evaluate our paper. We hope our detailed responses below can help address the reviewer's concerns.
>
> > W1: The model structure may be complex and resource-intensive. It is recommended to include a complexity and resource consumption analysis in both training and inference.
>
> **Ans**:
> Thank you for the suggestion. In Appendix Table A1 and Table A2, we provide analyses of the inference and training costs, along with potential directions for improvement. We plan to address these issues in future work to further enhance the practicality and applicability of JavisDiT.
>
> > W2: The paper lacks failure case analysis. It would be better to include typical failure cases and corresponding discussions, which would help reveal the model’s limitations and potential directions for improvement.
>
> **Ans**:
> A common failure case arises from the prompt-style gap. Since the text encoder in JavisDiT is frozen，the model may behave inconsistently when encountering prompts outside the training distribution during inference (e.g., highly complex captions). In the supplementary material under the `cmp_prompt_style/` directory, we compare JavisDiT’s behavior under raw prompts (aligned with the training style) and dense prompts (augmented with structured instructions), which exhibits two notable properties:
>
> 1. The learned spatio-temporal prior still generalizes effectively to certain keywords. For example, the phrase _glass clinking_ provides consistent priors for both the visual action (a spoon striking a glass) and the corresponding audio event, leading to synchronized audio–video generation.
>
> 2. The semantic encoder may lose information when the prompt becomes overly dense. For instance, under a dense prompt, JavisDiT completely ignores the clue _man speaking_, revealing limitations in instruction-following under high prompt complexity.
>
> In summary, a potential solution is to introduce an LLM-refinement stage that rewrites user prompts at deployment time into a style closer to the training distribution, thereby improving generalization. We plan to explore and validate this direction in future work.
>
> > W3: There is a slight overclaim in some parts of the writing. It is suggested to avoid overextended descriptions of timbre or speech-like attributes in the Introduction section to maintain precision and academic rigor.
>
> **Ans**:
> Thank you for the suggestion. Our initial use of “timbre” was intended to describe the audio source category (e.g., male, female, dog, car) rather than specific speech content. To avoid ambiguity, we have replaced it with the more precise term “source category” in the revised manuscript.
>
> > M1: The appendix contains comprehensive details. But, it may be better to move the ablation table 4 into the appendix while adding more details back to the main paper (eg, to further explain or validate the key design of HiST-Sypo).
>
> **Ans**: Thank you for the suggestion. Due to page limitations in the submission, we had to make certain compromises in the main content. After incorporating all revisions, we will reorganize the paper to improve readability and enrich the level of detail.
>
> > M2: Typo: '240P4s' in the Caption of Table A1 at Line 810 should be '240P 4s'?
>
> **Ans**:
> Thanks for pointing this out. We have updated it in the revised version.
>
>
> > Q1: An open question is whether it would be possible to use much less data to achieve comparable performance?
>
> **Ans**:
> This is indeed a valuable question. Based on our preliminary analysis and experiments, there appears to be a trade-off among diversity, quality, and quantity in the training data: for data-driven generative models, ensuring sufficient quantity is the first priority, after which quality and diversity can be gradually improved. We plan to investigate this relationship more thoroughly in future work.
>
> > Q2: Regarding the text semantic understanding module: In related works, the text encoder is often based on T5-like large language models, which can capture richer semantic representations. Has the author considered adopting similar LLM architectures to further enhance text semantic modeling capability?
>
> **Ans**:
> The text encoder we reuse from OpenSora is already based on the T5-XXL model. However, we do plan to explore stronger decoder-only LLMs or even multimodal LLMs as alternative text encoders, with the goal of leveraging their world knowledge and reasoning capabilities to better interpret user intent and further enhance instruction-following generation.

---

> > ### Comment · Reviewer_AE6y · 2025-11-27
> >
> > Thanks to the authors for your response. My concerns are well addressed. I also went through comments from other knowledgeable reviews. In general, I think the overall contribution in both method and experiments are solid. I would like to keep my original rating in the final stage and recommend its acceptance.

---

### Official Review · Reviewer_giZt · 2025-10-31

**Soundness:** 3
**Presentation:** 3
**Contribution:** 3
**Rating:** 8
**Confidence:** 4

**Summary:**

This paper addresses joint audio-video generation, focusing on two key objectives: (1) achieving strong generative performance for each individual modality, and (2) ensuring fine-grained synchronization between audio and video. To meet these goals, the authors propose JavisDiT, a diffusion transformer (DiT)-based model capable of generating high-quality audio and video. To enhance cross-modal alignment, they further introduce a hierarchical spatial-temporal synchronized prior estimation module. In addition, the paper presents a new benchmark, JavisBench, consisting of high-quality text-annotated sounding videos for evaluating joint audio-video generation.

**Strengths:**

1. The two identified challenges—developing a strong backbone for joint audio-video generation and achieving fine-grained audio-video synchronization—are well-motivated and reasonable.

2. Both the proposed JavisDiT model and the accompanying benchmark are valuable contributions that can provide meaningful insights to the research community.

3. The experimental evaluation is comprehensive and well-conducted.

**Weaknesses:**

1. Although the JavisDiT model is clearly described, its architectural design is not entirely intuitive. For example, the rationale behind the specific designs of the spatial-temporal self-attention and spatial-temporal cross-attention modules is not well explained. Providing more insights into the design choices would make the model structure easier to understand.

2. The proposed JavisScore is reasonable. It might be interesting to explore whether it could be extended in a hierarchical manner—for example, by incorporating multiple window sizes for evaluation.

3. Given the crucial role of the training dataset in determining both model accuracy and generalization, it would be helpful to clarify the main criteria used to ensure dataset validity. Additionally, it would be valuable to discuss how JavisDiT performs when potential biases arise from the heavy reliance on VLM-generated video captions.

**Questions:**

1. Training such a model requires substantial computational resources. Therefore, it would be valuable to further explore the broader potential of JavisDiT, for example, by examining its applicability to other related tasks.

2. The robustness of the model with respect to the quality of the training dataset should also be discussed, as dataset noise or imbalance could significantly affect performance and generalization.

---

> ### Author Response · Authors · 2025-11-26
> **Response to Reviewer giZt - Part I**
>
> We thank the reviewer for the valuable time and effort to evaluate our paper. We hope our detailed responses below can help address the reviewer's concerns.
>
> > W1: Although the JavisDiT model is clearly described, its architectural design is not entirely intuitive. For example, the rationale behind the specific designs of the spatial-temporal self-attention and spatial-temporal cross-attention modules is not well explained. Providing more insights into the design choices would make the model structure easier to understand.
>
> **Ans**:
> Thank you for the suggestion. We follow OpenSora in adopting spatial–temporal attention modules primarily to reduce computational cost. For video tokens of size ($T \times H \times W$), the computational complexity of full self-attention is
> $O\big((T H W)^2\big)$.
> In contrast, spatial–temporal attention reduces the complexity to
> $O\big(T (H W)^2 + T^2 (H W)\big) = O\big(T H W \cdot (T + H W)\big)$,
> which substantially lowers both memory consumption and computational overhead.
>
> We have added this discussion to Appendix C.1 of the revised manuscript.
>
>
> > W2: The proposed JavisScore is reasonable. It might be interesting to explore whether it could be extended in a hierarchical manner—for example, by incorporating multiple window sizes for evaluation.
>
> **Ans**:
> We attempted to integrate several combinations of window size and overlap that achieved the highest JavisScore accuracy in Figure A6, and evaluated these hierarchical schemes on JavisEval. However, as shown below, these combinations did not lead to noticeable performance improvements. A possible reason is that the current method’s similarity measure in the semantic space has already reached its accuracy limit, and aggregating evaluations across multiple window sizes may introduce additional noise, ultimately degrading performance. We plan to explore more advanced synchrony evaluation metrics in future work, potentially leveraging multimodal LLMs.
>
> - Experiments on hierarchical window size:
>
> | (window size, overlap) | ACC | AUROC |
> |---|---|---|
> | baseline: (2, 1.5) | **0.7514** | **0.6533** |
> | (1, 0.5) + (1.5, 1) + (2, 1.5) | 0.7346 | 0.6502 |
> | (1.5, 1) + (2, 1.5) + (2.5, 2) | 0.7408 | 0.6505 |
> | (2, 1.5) + (2.5, 2) + (3, 2.5) | 0.7427 | 0.6511 |
>
> - Experiments on hierarchical overlap:
>
> | (window size, overlap) | ACC | AUROC |
> |---|---|---|
> | baseline: (2, 1.5) | 0.7514 | 0.6533 |
> | (2, 0) + (2, 1) + (2, 1.5) | **0.7516** | **0.6534** |
> | (2, 0) + (2, 0.5) + (2, 1.5) | 0.7507 | 0.6518 |
> | (2, 0) + (2, 0.5) + (2, 1) + (2, 1.5) | 0.7508 | 0.6518 |
>
> > W3: Given the crucial role of the training dataset in determining both model accuracy and generalization, it would be helpful to clarify the main criteria used to ensure dataset validity. Additionally, it would be valuable to discuss how JavisDiT performs when potential biases arise from the heavy reliance on VLM-generated video captions.
>
> **Ans**:
> Thank you for the suggestion. We have added more details regarding the dataset validity as follows:
>
> - For diversity, our training data primarily combines TAVGBench with the multimodal sources from MMTrail. The millions of real video samples ensure substantial scene and category diversity. This is also reflected in the prompt distribution shown in Figure A2 of the revised manuscript.
>
> - For quality, we first use FunASR to filter out most videos containing speech. We then follow OpenSora and apply aesthetic filtering (threshold 4.0), motion filtering (threshold 0.1), and OCR filtering (threshold 5.0) to ensure a high-quality training set.
>
> On the other hand, since the training captions are generated by a VLM, we acknowledge that this may influence the real-world applicability of JavisDiT:
>
> - VLM-generated captions may contain hallucinations, resulting in inconsistencies between the input prompt and the generated video. While this could be mitigated by using a stronger captioning model or human annotation, such approaches would incur substantial cost.
>
> - The style of VLM-generated captions is fixed (e.g., simple high-level scene descriptions), whereas real user prompts may be extremely short (a single word) or highly complex (structured instructions). A common solution is to introduce an LLM-refinement stage to unify the prompt style, though this comes with additional inference overhead.

---

> > ### Author Response · Authors · 2025-11-26
> > **Response to Reviewer giZt - Part II**
> >
> > > Q1: Training such a model requires substantial computational resources. Therefore, it would be valuable to further explore the broader potential of JavisDiT, for example, by examining its applicability to other related tasks.
> >
> > **Ans**:
> > Thank you for the suggestion. We believe that JavisDiT has strong application potential in the following two scenarios:
> >
> > - Horizontal extension to other audiovisual generation tasks, such as video-to-audio generation (V2A), audio-to-video generation (A2V), and audio-driven image animation (AI2V). These tasks can be supported by designing appropriate masking strategies for reference and noise during inference. In the supplementary material (`app_to_avgen/`), we provide some preliminary explorations (e.g., a guitar-playing case). However, achieving effective applications will still require task-specific design and data preparation during training.
> >
> > - Vertical extension to more advanced multimodal tasks, such as instruction-based audiovisual editing and unified audiovisual understanding and generation. We plan to explore these directions further in future work.
> >
> > > Q2: The robustness of the model with respect to the quality of the training dataset should also be discussed, as dataset noise or imbalance could significantly affect performance and generalization.
> >
> > **Ans**:
> > As a data-driven method, the quality and diversity of the training data are indeed key factors that determine the performance upper bound of the model. Therefore, during dataset construction, we explicitly ensured both high quality and strong diversity (see our response to Weakness 3 for details) to mitigate potential generalization issues when deploying the trained model in real-world scenarios.

---

> > > ### Comment · Reviewer_giZt · 2025-11-27
> > >
> > > I believe the contribution of this paper is good, and the method is explained (after rebuttal) in a very clear way. I thus keep my rating unchanged as a "8: accept, good paper (poster)".

---

### Official Review · Reviewer_Fny1 · 2025-11-01

**Soundness:** 2
**Presentation:** 3
**Contribution:** 2
**Rating:** 4
**Confidence:** 4

**Summary:**

1. The paper proposes a joint audio-video generation model named JavisDiT, which is based on the Diffusion Transformer (DiT) architecture and aims to simultaneously generate high-quality and synchronized audio-video content from text prompts.
2. The paper designs a core module, the Hierarchical Spatial-Temporal Synchronized Prior Estimator, which extracts both global coarse-grained semantic and local fine-grained spatio-temporal priors from text. These priors are then injected into the DiT module to guide the precise spatial and temporal synchronization of the audio and video.
3. Addressing the lack of scene diversity and complexity in existing benchmarks, the authors constructed a new and more challenging benchmark dataset, JavisBench. If this dataset were to be made open-source, it would be beneficial to the community's development.
4. To address the deficiencies of existing metrics (such as AV-Align) , the paper introduces JavisScore. This new metric uses a temporal-aware semantic alignment mechanism to robustly measure synchronization in complex real-world scenarios.

**Strengths:**

The paper's motivation is clear and precisely targets the two core challenges in the field of Joint Audio-Video Generation (JAVG): ensuring high-quality generation of audio and video, and maintaining perfect synchronization between the two modalities. It is rare for a single work to contribute a model, dataset, and metric simultaneously. This represents a significant potential to advance the field and, as an academic work, is sufficiently compelling.

**Weaknesses:**

1. There is a clear disconnect between the paper's core claims and the qualitative results provided in the supplementary demo. The paper claims to solve high-quality generation and perfect synchronization, but the demos fail on both counts. The visual quality is significantly lower than the level of current video generation models, and audio-video synchronization is not well demonstrated in these demos, often appearing as coarse "scene-ambience" rather than the claimed "fine-grained" alignment. Furthermore, the number of samples is insufficient to cover the breadth of capabilities claimed. Critically, the demos lack the most essential evidence: direct comparisons against baseline methods (e.g., "T2V + V2A" or "T2A + A2V") for the same prompt. While the authors provide objective metrics in Table A8, the corresponding qualitative examples to visually substantiate these claims are missing.
2. The paper lacks sufficient subjective (human) evaluation. This is a necessary component for measuring video quality, audio quality, and audio-video alignment, and its absence is a significant omission.
3. The authors' justification for their model's weaker audio performance (e.g., in line 1538) is unconvincing. Attributing this weakness to supporting "variable-length audio generation" is a flawed argument, as the baseline AudioLDM 2 also supports this feature. Additionally, while the authors' concern about AudioLDM 2 training on AudioCaps is valid, their use of the term "data leakage" is imprecise.
4. The multi-stage training strategy is overly complex. Does the complex multi-stage training strategy require significant manual intervention and tuning when applied to new datasets.
5. The paper's readability is poor due to dense prose. For example, the term "Fine-Grained Spatial-Temporal Self-Attention Cross-Attention" (line 89) is structurally confusing. This confusion is amplified by the inconsistent naming in Figure 2 (e.g., "Fine-Grained ST-CrossAttn" and "ST-SelfAttn"), which hinders a clear understanding of the architecture. While the authors state in the Appendix (A.4) that LLMs were used "solely as writing assistants," we believe the paper's over-reliance on such tools has resulted in these convoluted descriptions and reduced overall readability.

**Questions:**

1. The HiST-Sypo estimator uses ImageBind's text encoder, while the proposed JavisScore metric also relies on ImageBind to measure audio-visual synchronization. Does this not create a 'circular' evaluation, where the model is effectively trained to optimize for the feature space of the metric that is supposed to be judging it?
2. Regarding the audio autoencoder, why did the authors opt for a Mel-spectrogram based VAE (which necessitates a vocoder) instead of a waveform-based VAE (e.g., Wave-VAE)? The choice of a Mel-VAE introduces an inherent quality loss from the vocoder, which could be a critical bottleneck for the final audio generation quality.
3. The paper claims to support variable-length audio generation, possibly via dynamic temporal masking. How is this practically implemented during inference? Given that the main evaluations are on 4-second videos, what evidence is there that this mechanism can robustly scale to longer durations (e.g., 10 seconds) while maintaining quality and synchronization?
4. In the ablation studies (Sec 5.3, line 795), the paper introduces normalized scores ($S_{AVQ}$, $S_{AVC}$, $S_{AVS}$) to simplify the results. What is the justification for the specific weighting factors used in these formulas? These equations appear arbitrary and lack a clear theoretical or empirical basis, making the ablation results difficult to interpret.

---

> ### Author Response · Authors · 2025-11-26
> **Response to Reviewer Fny1 - Part I**
>
> We thank the reviewer for the valuable time and effort to evaluate our paper. We hope our detailed responses below can help address the reviewer's concerns.
>
> > W1: There is a clear disconnect between the paper's core claims and the qualitative results provided in the supplementary demo. The paper claims to solve high-quality generation and perfect synchronization, but the demos fail on both counts. Critically, the demos lack the most essential evidence: direct comparisons against baseline methods (e.g., "T2V + V2A" or "T2A + A2V") for the same prompt.
>
> **Ans**:
> First, regarding quality and synchrony, most of the previously shown cases in the supplementary material were 240p-resolution examples (used for objective evaluation), where the visual quality and synchrony may not be very noticeable. In the updated supplementary material, we now provide more high-resolution cases (480p–720p) as well as clearer audiovisual event cues beyond simple ambience in the `hr_demos/` folder.
>
> Second, we would like to clarify that “perfect synchronization” is our goal, not the current capability of the model, and we did not claim this in the contribution part. Despite this, we have replaced such terms with “pursuing precise synchronization” to avoid ambiguity and ensure rigorousness.
>
> Finally, we also include comparisons with the sequential baselines in the `cmp_baseline/` folder of the supplementary material, which more clearly demonstrate the advantages of our method.
>
>
> > W2: The paper lacks sufficient subjective (human) evaluation. This is a necessary component for measuring video quality, audio quality, and audio-video alignment, and its absence is a significant omission.
>
> **Ans**:
> Thanks for this suggestion.
> Here, we randomly sample 100 prompts from JavisBench and run UniVerse-1 and our JavisDiT model to generate corresponding audio–video outputs. Three volunteers are then recruited to perform blind win–tie–lose
> preference judgments on three aspects: video quality, audio quality, and audio-video alignment. The averaged scores across annotators are used as the final evaluation metric.
>
> | Aspect | JavisDiT-win | JavisDiT-tie | JavisDiT-lose |
> |---|:---:|:---:|:---:|
> | Video Quality | 24.7% | 20.0% | 55.3% |
> | Audio Quality | 55.3% | 26.7% | 18.0% |
> | A-V Alignment | 38.6% | 56.0% |  5.4% |
>
> The human evaluation results are consistent with the objective metrics reported in our manuscript:
>
> 1. Our video quality is indeed slightly lower than that of UniVerse-1. This mainly stems from the performance gap between the backbones: our method currently relies on the OpenSora-1.2 backbone, which is notably weaker than the Wan-2.1 backbone used in UniVerse-1. We expect this issue to be substantially alleviated as stronger T2V backbones become available in future versions.
>
> 2. Our audio quality and audio–video alignment outperform UniVerse-1. This is because UniVerse-1 adopts a relatively complex architecture to bridge two separately pretrained branches, and the resulting representation gap weakens cross-modal interaction. In contrast, our JavisDiT design offers a more streamlined and effective mechanism for enabling tight audio–video coupling.
>
> We haved added the experiments and analysis in Section 5.4 in the revised version.
>
>
> > W3: The authors' justification for their model's weaker audio performance (e.g., in line 1538) is unconvincing. Attributing this weakness to supporting "variable-length audio generation" is a flawed argument, as the baseline AudioLDM 2 also supports this feature. Additionally, while the authors' concern about AudioLDM 2 training on AudioCaps is valid, their use of the term "data leakage" is imprecise.
>
> **Ans**:
> First, we would like to clarify that AudioLDM2 is trained exclusively on 10.24-second audio. Although one can forcibly alter the generation length during inference by modifying the shape of the noisy latent (e.g., changing it to 4 seconds), the model’s denoising process becomes misaligned with its training distribution, leading to a significant drop in generation quality, as shown in the table below.
>
> In contrast, since our audio pretraining is performed under variable-length settings, the generation quality at 4 seconds remains much closer to that of AudioLDM2. We believe that collecting more data and training for additional iterations will further improve generation performance across different lengths. The corresponding experiments and discussion have been added to Appendix E.1.
>
> | Model | AudioCaps-10s |  |  | AudioCaps-4s |  |  |
> |---|---|---|---|---|---|---|
> |  | FAD↓ | TA-IB↑ | CLAP↑ | FAD↓ | TA-IB↑ | CLAP↑ |
> | AudioLDM2 | 2.01 | 0.205 | 0.487 | 5.57 | 0.147 | 0.326 |
> | JavisDiT | 5.19 | 0.164 | 0.356 | 6.23 | 0.141 | 0.301 |
>
> Additionally, we agree with the reviewer’s comment and have replaced the inappropriate term “data leakage” with “overfitting,” along with the corresponding revisions in the paper.

---

> > ### Author Response · Authors · 2025-11-26
> > **Response to Reviewer Fny1 - Part II**
> >
> > > W4: The multi-stage training strategy is overly complex. Does the complex multi-stage training strategy require significant manual intervention and tuning when applied to new datasets.
> >
> > **Ans**:
> > The multi-stage paradigm is a common approach for building foundation models (e.g., T2V base models[1][2]), enabling the model to progressively acquire more complex capabilities. This is aligned with our motivation in extending a T2V backbone to the T2AV JavisDiT framework, and we believe it is a reasonable and well-established design choice.
> >
> > More importantly, when transferring to a new dataset, the entire data-processing pipeline can be reused without manual intervention, which greatly enhances the model’s generalization and adaptability across different scenarios.
> >
> > > W5: The paper's readability is poor due to dense prose. For example, the term "Fine-Grained Spatial-Temporal Self-Attention Cross-Attention" is structurally confusing.
> >
> > **Ans**:
> > Thank you for pointing this out. This was indeed a typo. The intention is to describe the fine-grained cross-attention guided by the spatiotemporal prior; therefore, the module should be named _Fine-Grained Spatial-Temporal Cross-Attention (ST-CrossAttn)_, consistent with the naming in Figure 2. We have corrected this in the revised manuscript.
> >
> > > Q1: The HiST-Sypo estimator uses ImageBind's text encoder, while the proposed JavisScore metric also relies on ImageBind to measure audio-visual synchronization. Does this not create a 'circular' evaluation, where the model is effectively trained to optimize for the feature space of the metric that is supposed to be judging it?
> >
> > **Ans**:
> > We would like to clarify that the two components use entirely different encoders, and, importantly, they operate in different representation spaces. The ST-Prior estimator includes the text-encoder with an additional transformer module that projects the features into a new space, whereas JavisScore remains entirely operates visual- and audio-encoder within the original ImageBind feature space. Therefore, there is no overlap between the two spaces, and no issue of circular evaluation arises.
> >
> > > Q2: Regarding the audio autoencoder, why did the authors opt for a Mel-spectrogram based VAE (which necessitates a vocoder) instead of a waveform-based VAE (e.g., Wave-VAE)?
> >
> > **Ans**:
> > There are two main reasons for this choice:
> > 1. Most modern audio generation models use Mel-spectrogram VAEs[3][4][5], as they are efficient and incur minimal information loss.
> > 2. Since the T2A branch is warm-started from a pretrained T2V model, and a Mel-spectrogram is essentially an image-like representation, it can effectively inherit the pretrained visual knowledge.
> >
> > > Q3: The paper claims to support variable-length audio generation, possibly via dynamic temporal masking. How is this practically implemented during inference? What evidence is there that this mechanism can robustly scale to longer durations (e.g., 10 seconds) while maintaining quality and synchronization?
> >
> > **Ans**:
> > Specifically, our variable-length generation is not achieved through temporal masking; instead, we directly modify the (temporal) shape of the noise latent and apply the subsequent denoising process accordingly.
> >
> > We have extended the evaluation on 10-second generations using JavisBench-mini to provide a more comprehensive comparison. The results show that our method maintains comparable quality (FVD, FAD), consistency (CLIP, CLAP, AVHScore), and synchrony (JavisScore).
> >
> > | Length | FAD↓ | FVD↓ | CLIP↑ | CLAP↑ | AVHScore↑ | JavisScore↑ |
> > |---|:---:|:---:|:---:|:---:|:---:|:---:|
> > | 4s | 241.8 | 7.3 | 0.308 | 0.382 | 0.186 | 0.153 |
> > | 10s | 233.8 | 7.1 | 0.307 | 0.385 | 0.183 | 0.154 |
> >
> > We have incorporated the corresponding experiments and discussion into Section 5.3 of the revised manuscript.
> >
> > > Q4: In the ablation studies, the paper introduces normalized scores to simplify the results. What is the justification for the specific weighting factors used in these formulas? These equations appear arbitrary and lack a clear theoretical or empirical basis, making the ablation results difficult to interpret.
> >
> > **Ans**:
> > Our purpose in choosing these weighting factors is simply to bring the different metrics onto a comparable scale, making detailed comparison more convenient. The complete results have been added to Section E.1 of the appendix.
> >
> >
> > ---
> >
> > [1] Zangwei Z, Xiangyu P, et al. Open-sora: Democratizing efficient video production for all. ArXiv, 2024.
> >
> > [2] Wan Team. Wan: Open and advanced large-scale video generative models. ArXiv, 2024.
> >
> > [3] Liu H, Yuan Y, et al. Audioldm 2: Learning holistic audio generation with self-supervised pretraining. IEEE TASLP, 2024.
> >
> > [4] Zhang Y, Gu Y, et al. Foleycrafter: Bring silent videos to life with lifelike and synchronized sounds. ArXiv, 2024.
> >
> > [5] Cheng H K, Ishii M, et al. MMAudio: Taming Multimodal Joint Training for High-Quality Video-to-Audio Synthesis. CVPR, 2025.

---

### Official Review · Reviewer_nc3c · 2025-11-02

**Soundness:** 3
**Presentation:** 3
**Contribution:** 3
**Rating:** 6
**Confidence:** 4

**Summary:**

JavisDiT is an end-to-end Diffusion Transformer ($\text{DiT}$) architecture that simultaneously generates high-quality video and audio content. The core innovation is the Hierarchical Spatial-Temporal Synchronized Prior ($\text{HiST-Sypo}$) Estimator. This module extracts both coarse-grained semantic priors (overall framework) and fine-grained spatio-temporal priors (timing and location of events) from the input text to guide the synchronization of the generated video and audio.

Key Contributions:

Novel JAVG Architecture (JavisDiT): A unified $\text{DiT}$-based system that leverages spatio-temporal self-attention, cross-attention, and a Multi-Modality Bidirectional Cross-Attention ($\text{MM-BiCrossAttn}$) module to achieve high-quality joint generation.

Hierarchical Synchronization Mechanism ($\text{HiST-Sypo}$): A fine-grained alignment module that estimates and injects text-derived spatial priors (where an event occurs) and temporal priors (when an event starts/stops) into the generation process.

New Benchmark ($\text{JavisBench}$) and Metric ($\text{JavisScore}$): The paper introduces $\text{JavisBench}$ ($\sim$10,140 samples), a challenging dataset focused on complex, multi-event, real-world scenarios (including sequential and simultaneous events). It also proposes $\text{JavisScore}$, a temporal-aware semantic metric designed to more robustly evaluate fine-grained synchronization than previous methods.

**Strengths:**

The combination of $\text{DiT}$ as a backbone for joint audio-video generation with the Hierarchical Spatio-Temporal Prior ($\text{HiST-Sypo}$) Estimator is highly novel. Previous JAVG methods lacked explicit fine-grained spatio-temporal modeling. The $\text{JavisScore}$ metric, which targets the least synchronized frames to increase sensitivity to local desynchronization, is also an original contribution to evaluation methodology.

The system design, including the detailed module structure (Figure 2) and the three-stage training strategy (pretraining, $\text{ST-Prior}$ training with contrastive loss, $\text{JAVG}$ training), is robust. The ablations demonstrate that the $\text{HiST-Sypo}$ estimator provides a significant and consistent gain in $\text{AV-Consistency}$ and $\text{AV-Synchrony}$ compared to simple bidirectional cross-attention ($\text{BICA}$). Furthermore, the empirical verification confirms that $\text{JavisScore}$ is substantially more reliable ($\sim$23% higher accuracy) than $\text{AV-Align}$ in distinguishing synchronous from asynchronous pairs.

The paper clearly articulates the dual challenges of high-quality generation and fine-grained synchronization. The distinction between $\text{Spatial Alignment}$ and $\text{Temporal Alignment}$ is precisely defined. The framework is well-diagrammed, and the appendix provides ample technical detail on data curation, augmentation for negative samples, and loss functions.

This work significantly elevates the bar for $\text{JAVG}$ research. $\text{JavisDiT}$ achieves $\text{SOTA}$ performance across both established and the new, challenging $\text{JavisBench}$ dataset, setting a new standard for generation quality and synchronization coherence. The introduction of $\text{JavisBench}$, focused on multi-event complexity (75% multiple events, 57% simultaneous events), directly addresses the "out-of-domain" issue that hinders the practical applicability of current models.

**Weaknesses:**

Computational Cost and Efficiency: The paper acknowledges the high computational overhead of the $\text{DiT}$-based architecture and diffusion process9. The latency analysis in Table $\text{A1}$ indicates that generating a small $240\text{P}, 4\text{s}$ clip takes $\mathbf{30}$ seconds on an $\text{H}100 \text{ GPU}$, which is prohibitive for real-time or fast creative applications. The training of the full JAVG model takes $\mathbf{256}$ GPU days on $\text{H}100$10. Suggestion: Discuss explicit strategies for accelerating inference (e.g., consistency models or distilled sampling methods) as a crucial next step to transition the model from research to practical use.

Dataset Leakage and Bias: While the authors took steps to avoid leakage by collecting videos between June and December 2024 and filtering 11, they trained the audio branch on vast public datasets ($\sim 788\text{K}$ entries) including $\text{AudioSet}$ and $\text{VGGSound}$12. These datasets are known to contain biases (e.g., $\text{off-screen}$ sounds) and may overlap non-trivially with the test sets of cascaded models (which often use V2A models trained on these very datasets). Suggestion: Explicitly discuss how the $\text{HiST-Sypo}$ training (Stage 2) mitigates biases learned by the audio branch during pretraining (Stage 1).

Ambiguity in Prior Estimation: The fine-grained $\text{ST-Prior}$ estimation adopts a $\text{VAE}$-like sampling strategy to model the variability of events for the same text prompt: $(p_{s},p_{t})\leftarrow\mathcal{P}_{\phi}(s;\epsilon)$13131313. However, it's unclear how much of the final generation is controlled by the sampled noise ($\epsilon$) vs. the core semantics. Suggestion: Discuss or ablate the influence of the sampling variance ($\mathcal{P}_{\phi}$'s variance output) on the final generated synchronization, specifically for multi-event or off-sc

**Questions:**

Robustness to New Language (Zero-Shot/Few-Shot): The HiST-Sypo relies on an ImageBind encoder trained on text-image pairs. How robust is the HiST-Sypo estimator to novel descriptive phrases (e.g., neologisms, highly contextual language) that were not present in its training corpus? Does the learned spatio-temporal prior generalize effectively when the text is complex and abstract?

Necessity of Two-Stage Prior Training: The ST-Prior Estimator is trained separately (Stage 2) using a complex contrastive learning pipeline (four loss functions, negative sampling strategies). Was a simpler end-to-end training strategy (e.g., training the estimator jointly with the full JAVG model using only the JAVG loss and minimizing a simpler feature distance) attempted? If so, why did it perform worse than this multi-stage approach?

Visualization of Fine-Grained Alignment: Figure 7 is a compelling visualization of the Spatial and Temporal Attention maps, confirming the model focuses on the correct object ("bubbles" vs. "diver"). Could the authors provide a similar visualization for a multi-event, sequential or simultaneous case (e.g., like the one in Figure 1, robot and dog tussle, then aliens talk) to demonstrate how the attention maps shift over time to capture the different spatial and temporal priors injected?

JavisScore Parameter Selection Justification: The authors selected Topk-min-40% as the synchronization estimation strategy (focusing on the 40% least synchronized frames). The ablation (Fig. A6) shows only minor variations in AUROC across most Topk thresholds. Why was 40% chosen over 50% or 60%? Is there a theoretical or perceptual reason for focusing on the hardest 40% of the clip, and did human evaluation confirm that desynchronization is most noticeable at these specific thresholds?

---

> ### Author Response · Authors · 2025-11-26
> **Response to Reviewer nc3c - Part I**
>
> We thank the reviewer for the valuable time and effort to evaluate our paper. We hope our detailed responses below can help address the reviewer's concerns.
>
> > W1: High computational cost and inefficiency in both training and inference stages. Discussing explicit strategies for accelerating inference is a crucial next step to transition the model from research to practical use.
>
> **Ans**:
> Thanks for the suggestion. Based on our analysis in Table A1, the model can be further accelerated along several concrete directions:
>
> - _Module level_: Injecting the ST-Prior introduces additional blocks, parameters, and computation, since both the video and audio branches must be modulated simultaneously. Therefore, developing a more lightweight module design is a highly promising direction.
>
> - _Modality level_: The video branch consumes substantially more resources than the audio branch due to the much larger number of video tokens. Thus, exploring video-token compression could yield significant efficiency gains.
>
> - _Model level_: Reducing model size or sampling steps through distillation is another direct way to improve overall inference efficiency.
>
> - _Application level_: Framework-level and hardware-specific optimizations can also provide additional, case-by-case acceleration.
>
> We will continue to explore more efficient strategies to improve the deployment of our JavisDiT model.
>
> > W2: Dataset Leakage and Bias: The authors trained the audio branch on vast public datasets (eg, AudioSet and VGGSound) that contain biases (e.g., off-screen sounds) and may overlap non-trivially with the test sets of cascaded models (which often use V2A models trained on these very datasets).
>
> **Ans**: We hope to clarify that:
> 1. **Regarding dataset leakage**: We would like to clarify that the datasets used during audio pretraining are the training splits of AudioSet and VGGSound, which do not overlap with the test sets of the cascaded models (e.g., the V2A model). Therefore, no dataset leakage occurs.
>
> 2. **Regarding dataset bias**: Although the audio branch is initialized from large-scale audio-only pretraining, the subsequent joint audio–video training introduces and updates a substantially larger set of parameters. Moreover, optimizing the synchronized loss conditioned on the spatiotemporal prior encourages the audio branch to learn richer on-screen actions and visual dynamics, thereby suppressing off-screen or scene-agnostic patterns inherited from the audio-only pretraining.
>
> > W3: Ambiguity in Prior Estimation: The fine-grained ST-Prior estimation adopts a VAE-like sampling strategy to model the variability of events for the same text prompt; however, it is not clear how much of the final generation is controlled by the sampled noise vs the core semantics.
>
> **Ans**:
> Thank you for the suggestion. We evaluated how different sampling variances affect the final audio–video synchrony (JavisScore) when using the ST-Prior. The results are shown below:
>
> | Variance | Average | Multi-Event | Off-Screen |
> |---|:---:|:---:|:---:|
> | 0.0 * $\sigma^2$ | 0.1515 | 0.1527 | 0.1424 |
> | 0.5 * $\sigma^2$ | 0.1517 | 0.1526 | 0.1363 |
> | **1.0 * $\sigma^2$** | **0.1530** | **0.1537** | **0.1502** |
> | 2.0 * $\sigma^2$ | 0.1501 | 0.1517 | 0.1335 |
>
> Since our ST-Prior models a distribution rather than a deterministic semantic embedding (both the mean and variance are predicted), manually altering the sampling variance (either increasing it by 2× or decreasing it by 0.5×) disrupts the learned distribution. Consequently, this leads to a degradation in audio–video synchrony in the generated outputs.

---

> > ### Author Response · Authors · 2025-11-26
> > **Response to Reviewer nc3c - Part II**
> >
> > > Q1: Robustness to New Language: How robust is the HiST-Sypo estimator to novel descriptive phrases (e.g., neologisms, highly contextual language) that were not present in its training corpus? Does the learned spatio-temporal prior generalize effectively when the text is complex and abstract?
> >
> > **Ans**:
> > The prompt style indeed has an impact on prior generalization. Since we freeze the ImageBind text encoder, the model may behave inconsistently when encountering prompts outside the training distribution during inference (e.g., highly complex captions). In the supplementary material under the `cmp_prompt_style/` directory, we compare JavisDiT’s behavior under raw prompts (aligned with the training style) and dense prompts (augmented with structured instructions). The HiST-Sypo estimator exhibits two notable properties:
> >
> > 1. The learned spatio-temporal prior still generalizes effectively to certain keywords. For example, the phrase _glass clinking_ provides consistent priors for both the visual action (a spoon striking a glass) and the corresponding audio event, leading to synchronized audio–video generation.
> >
> > 2. The semantic encoder may lose information when the prompt becomes overly dense. For instance, under a dense prompt, JavisDiT completely ignores the clue _man speaking_, revealing limitations in instruction-following under high prompt complexity.
> >
> > In summary, a potential solution is to introduce an LLM-refinement stage that rewrites user prompts at deployment time into a style closer to the training distribution, thereby improving generalization. We plan to explore and validate this direction in future work.
> >
> >
> > > Q2: Necessity of Two-Stage Prior Training: The ST-Prior Estimator is trained separately (Stage 2) using a complex contrastive learning pipeline. Was a simpler end-to-end training strategy attempted? If so, why did it perform worse than this multi-stage approach?
> >
> > **Ans**:
> > The key observation is that the contrastive learning introduces new information (or knowledge) by extracting additional spatiotemporal priors from the user’s raw prompt to guide sounding-video generation. If Stage 2 were removed and the prior estimator were trained jointly with the JAVG-DiT from scratch, both components would be fitted on the same distribution constrained by the training dataset, making the presence or absence of the prior estimator essentially equivalent. However, the results in Table 3 show that removing the prior estimator leads to a significant performance drop, indicating that it is necessary to maintain a multi-stage training pipeline in order to independently extract meaningful spatiotemporal priors.
> >
> > > Q3: Visualization of Fine-Grained Alignment: Could the authors provide a similar visualization for a multi-event, sequential or simultaneous case to demonstrate how the attention maps shift over time to capture the different spatial and temporal priors injected?
> >
> > **Ans**:
> > Thanks for this constructive suggestion. We include a more detailed example in the Appendix to illustrate how attention shifts when two cats meow alternately:
> >
> > - First, when the orange cat on the left meows, the visual spatial attention concentrates on its head, which corresponds to the region highlighted by the white dashed box in the audio-attention map.
> >
> > - Then, when the tabby cat on the right meows (indicated by the red dashed box in the audio spectrogram), the visual attention shifts to the tabby’s head, providing a more direct spatial prior.
> >
> > - Finally, when the orange cat meows again, both spatial and temporal attention shift once more, injecting updated prior information into the audio and video branches.
> >
> > This example more clearly demonstrates how spatiotemporal priors are injected and shift over sequential events, further validating the effectiveness of our method.
> >
> > > Q4: JavisScore Parameter Selection Justification: Why was 40% chosen over 50% or 60%? Is there a theoretical or perceptual reason or human evaluation?
> >
> > **Ans**:
> > The parameter choice in JavisScore is empirical. The 40% threshold does not have a strict theoretical justification; rather, it was selected based on the relatively better hyperparameter search results shown in Figure A6. We also manually verified on the JavisEval dataset that the desynchronization estimates obtained under this threshold are indeed more accurate.

---

### Author Response · Authors · 2025-12-01
**Rebuttal Summary**

Dear ACs, SACs, and PCs,

We thank the reviewers for their time and constructive discussion, and also feel sorry for the impact caused by the recent OpenReview bug. Below, we provide a concise summary of the reviewers’ initial concerns and the outcomes of the rebuttal and discussion phase, and hope this summary helps the AC’s effort in making the final decision.

### Initial Concerns

The reviewers’ initial concerns centered on the following aspects:

1. **Claim justification**:
Reviewers Fny1 and AE6y raised questions about several claim statements. In response, we carefully clarified each point, revised overly strong wording, and added additional qualitative and quantitative results to further substantiate our core contributions.
2. **Lack of additional experiments**:
Reviewers requested further ablations (e.g., sampling noise, generation length, metric extensions) and human evaluations against baselines. We provided point-by-point responses and added new experiments to directly address these concerns.
3. **Needs of further discussion**:
The reviewers offered suggestions regarding training and inference efficiency, prompt-style transfer, dataset quality and diversity, etc. We provided detailed explanations and included corresponding discussion and preliminary exploration results in the revised manuscript and supplementary material.

### Rebuttal and Discussion Progress

During the rebuttal phase, we provided extensive experiments and detailed explanations that addressed the reviewers’ concerns. Before the large-scale leakage on Nov. 27, we had already received explicit positive feedback:

- Reviewer giZt: “I believe the contribution of this paper is good, and the method is explained (after rebuttal) in a very clear way.”
**Maintained the rating of 8, ~12 hours before the bug.**
- Reviewer AE6y: “I also went through comments from other knowledgeable reviews. In general, I think the overall contribution in both method and experiments are solid.”
**Maintained the rating of 8, ~9 hours before the bug.**

Although reviewer Fny1 (initial rating 4) and nc3c (initial rating 6) did not respond before the discussion cut-off, we believe the substantial new experimental evidence and thorough clarifications are sufficient to address their remaining concerns:

- Reviewer Fny1’s main concerns focused on the **rigor of the paper’s claims**, including (1) supplementary demos, (2) justification of audio performance, and (3) variable-length generation. In response, we provided **corresponding qualitative and quantitative experimental results as evidence and revised several statements** to ensure greater precision in the claims. Additional concerns, such as the need for human evaluation and improved presentation clarity, were also addressed **point-by-point**.
- Reviewer nc3c raised several **open questions** regarding model efficiency, dataset bias, prior estimation, and the need for further visualization and justification. We provided **qualitative and quantitative results along with detailed discussions**, all of which have been incorporated into the revised manuscript.

### Conclusion

The reviewers consistently recognize that **our work is well-motivated, the model design is novel and robust, and the empirical experiments, along with the proposed benchmark and metric, provide valuable standards and insights for future research in the field.** All new experiments, discussions, and clarifications have been incorporated into the revised manuscript.

We once again thank the reviewers and AC for their efforts, and we hope this summary is helpful for the final decision.

Sincerely,

Authors

---

### Meta-Review · Area_Chair_9r7F · 2026-01-08

**Summary:**

This paper proposes JavisDiT, an end-to-end joint audio-video diffusion transformer utilizing a Hierarchical Spatio-Temporal Synchronized Prior (HiST-Sypo) to achieve fine-grained synchronization. To support this, we introduce JavisBench, a large-scale multi-event benchmark, and JavisScore, a dedicated synchronization metric. Empirical results show that JavisDiT outperforms existing cascaded and joint methods, delivering superior spatio-temporal alignment and competitive generation quality.

**Reviewer Concerns:**

+ concerns addressed by the rebuttal
a. claims and overstatement
Strong claims (e.g., “perfect synchronization”) were revised and softened. Additional qualitative examples and quantitative results were added. Terminology such as “data leakage” was corrected to more precise wording (“overfitting”).
b. missing qualitative evidence and baseline (Fny1)
High-resolution demos (480p–720p) were added. Direct comparisons with sequential baselines (T2V+V2A, T2A+V2A) were included in supplementary materials. Qualitative gaps identified by the reviewer were explicitly addressed.
c. human evaluation
A blind human study (100 prompts, win–tie–lose) was added, evaluating video quality, audio quality, and A–V alignment. Human results were consistent with objective metrics (weaker video quality, stronger audio quality and alignment).
d. data leakage and bias
Authors clarified dataset splits and timing (YouTube videos collected after benchmark release). They explained how joint training and synchronization losses mitigate audio-only pretraining bias.
e. visualization of fine-grained synchronization
Additional multi-event attention visualizations were added (sequential events, alternating sound sources), clarifying how spatial and temporal priors shift over time.

-concerns still outstanding
a. computational cost and practicality
Training and inference costs remain very high (e.g., ~30s for 4s clip on H100). The rebuttal discusses future directions (distillation, compression), but no concrete acceleration is demonstrated. This remains a legitimate concern for real-world deployment.
b. dependence on backbone strength
Video quality lags behind UniVerse-1 due to weaker T2V backbone (OpenSora vs. Wan-2.1). While acknowledged and reasonably justified, this limits the perceived competitiveness of generation quality.
c. metric design choice (JavisScore 40% threshold)
lacks a strong theoretical or perceptual justification. Human verification was mentioned but remains limited.

**Reviewer Scores:**

giZt -8
nc3c - 6
Fny1 - 4
Majority postive to borderline

---

### Decision · Program_Chairs · 2026-01-26

Accept (Poster)